# Lactate-mediated epigenetic reprogramming regulates formation of human pancreatic cancer-associated fibroblasts

Tushar D Bhagat[1], Dagny Von Ahrens[1], Meelad Dawlaty[1], Yiyu Zou[1], Joelle Baddour[2], Abhinav Achreja[2], Hongyun Zhao[2], Lifeng Yang[2], Brijesh Patel[3], Changsoo Kwak[4,5], Gaurav S Choudhary[1], Shanisha Gordon-Mitchell[1], Srinivas Aluri[1], Sanchari Bhattacharyya[1], Srabani Sahu[1], Prafulla Bhagat[1], Yiting Yu[1], Matthias Bartenstein[1], Orsi Giricz[1], Masako Suzuki[1], Davendra Sohal[6], Sonal Gupta[4,5], Paola A Guerrero[4,5], Surinder Batra[7], Michael Goggins[8], Ulrich Steidl[1], John Greally[1], Beamon Agarwal[9], Kith Pradhan[1], Debabrata Banerjee[3], Deepak Nagrath[10]*, Anirban Maitra[4,5], Amit Verma[1]*

[1]Albert Einstein College of Medicine, Montefiore Medical Center, New York, United States; [2]Department of Biomedical Engineering, University of Michigan, Ann Arbor, United States; [3]Rutgers University, New Brunswick, United States; [4]Department of Pathology, Sheikh Ahmed Pancreatic Cancer Research Center, UT MD Anderson Cancer Center, Houston, United States; [5]Department of Translational Molecular Pathology, Sheikh Ahmed Pancreatic Cancer Research Center, UT MD Anderson Cancer Center, Houston, United States; [6]Department of Medicine, Cleveland Clinic, Cleveland, United States; [7]University of Nebraska Medical Center, Omaha, United States; [8]Johns Hopkins, Baltimore, United States; [9]GenomeRxUs LLC, Secane, United States; [10]Biointerfaces Institute, University of Michigan, Ann Arbor, United States

*For correspondence:
dnagrath@umich.edu (DN);
AMaitra@mdanderson.org (AM);
amit.verma@einstein.yu.edu (AV)

**Competing interests:** The authors declare that no competing interests exist.

**Abstract** Even though pancreatic ductal adenocarcinoma (PDAC) is associated with fibrotic stroma, the molecular pathways regulating the formation of cancer associated fibroblasts (CAFs) are not well elucidated. An epigenomic analysis of patient-derived and de-novo generated CAFs demonstrated widespread loss of cytosine methylation that was associated with overexpression of various inflammatory transcripts including *CXCR4*. Co-culture of neoplastic cells with CAFs led to increased invasiveness that was abrogated by inhibition of *CXCR4*. Metabolite tracing revealed that lactate produced by neoplastic cells leads to increased production of alpha-ketoglutarate (aKG) within mesenchymal stem cells (MSCs). In turn, aKG mediated activation of the demethylase TET enzyme led to decreased cytosine methylation and increased hydroxymethylation during de novo differentiation of MSCs to CAF. Co-injection of neoplastic cells with TET-deficient MSCs inhibited tumor growth in vivo. Thus, in PDAC, a tumor-mediated lactate flux is associated with widespread epigenomic reprogramming that is seen during CAF formation.
DOI: https://doi.org/10.7554/eLife.50663.001

## Introduction

Pancreatic ductal adenocarcinoma (PDAC) is a deadly disease and is the third leading cause of deaths from cancer in the United States. An exuberant host fibrotic response, termed stromal

desmoplasia, is a characteristic feature of PDAC (*Feig et al., 2013*; *Biffi et al., 2019*), (*Feig et al., 2012*; *Yu et al., 2012*). The stromal fibrosis is suspected to contribute to chemoresistance in pancreatic cancer by impeding drug delivery (*Provenzano et al., 2012*; *Öhlund et al., 2017*; *von Ahrens et al., 2017*), and differences in stromal behavior have been implicated with patient outcomes in pancreatic and other cancers. The stromal microenvironment predominantly consists of cancer-associated fibroblasts (CAFs) that are activated during tumorigenesis, undergoing morphological and functional changes, when compared to normal fibroblasts (*Kalluri, 2016*). CAFs are derived via activation of resident pancreatic stellate cells, and also from differentiation and activation of bone marrow derived mesenchymal stem cells (BM-MSCs) that migrate to the peritumoral milieu due to chemotactic signals released by cancer cells. The characteristic features of activated CAFs include the expression of α-smooth-muscle actin (α-SMA (*ACTA1*)) (*Öhlund et al., 2017*), enhanced secretory and contractile ability, increased synthesis of extracellular matrix proteins, such as collagens, and of growth factors including basic fibroblast growth factor, transforming growth factor beta (TGF-β), interleukin-8 and platelet-derived growth factors (PDGF). In preclinical models of PDAC (*Feig et al., 2012*; *Hwang et al., 2008*; *Xu et al., 2010*), as well as in other cancer types (*Karnoub et al., 2007*), CAFs have been shown to promote invasion and metastases through myriad mechanisms. Multiple lines of evidence support the existence of robust paracrine signals from neoplastic epithelial cells to the stromal compartment (*Behrens et al., 2010*; *Bailey et al., 2009*; *Omary et al., 2007*), which likely facilitates the reprogramming of MSCs to an activated CAF-like state, that in turn, promotes PDAC progression. Careful studies using high density copy number arrays and mutational profiling have excluded the presence of genomic alterations in pancreatic CAFs, potentially suggesting that reprogramming is most likely epigenetic in nature (*Walter et al., 2008*). While the epigenome of the PDAC neoplastic epithelium has been extensively studied (*Sato and Goggins, 2006*; *Goggins, 2005*), the stromal epigenome is largely uncharacterized. Thus, the underlying objective of this study was to study patterns of epigenetic reprogramming in the PDAC stroma, specifically the most predominant α-SMA (*ACTA1*) expressing CAF cell type (*Öhlund et al., 2017*).

In the present study, we studied genome wide cytosine methylation in CAFs by using both primary CAFs derived from resected PDAC, as well as de novo CAFs generated from MSCs in vitro. Our analysis revealed widespread loss of DNA methylation in CAFs as the dominant 'epi-genotype'. This epigenetic reprogramming was associated with upregulation of numerous transcripts, including those encoding the chemokine receptor *CXCR4*. Our data reveal that stromal *CXCR4* overexpression promotes PDAC invasion, and provides a facile druggable target within the tumor microenvironment attenuating tumor progression. Importantly, from a mechanistic standpoint, we determine that paracrine lactate secreted by PDAC cells can be incorporated in stromal cells and lead to increased alpha-keto glutarate (aKG). This is associated with activation of the TET demethylase, thus potentially leading to epigenetic reprogramming seen during CAF formation. Our studies underscore the emerging thread between aberrant metabolism and epigenomic alterations in cancer progression, albeit from the aspect of peritumoral stroma in PDAC.

## Results

### Widespread epigenetic reprogramming is observed in primary and de novo transformed CAFs

Primary cultures of cancer-associated fibroblasts (CAFs) were established from seven surgically resected PDAC tissue samples and used for epigenomic and transcriptomic analysis. Genome wide cytosine methylation was performed by the *HpaII* tiny fragment Enrichment by Ligation-mediated PCR (HELP) assay that relies on differential digestion by *HpaII* and *MspI* to identify methylated CpG sites (*Figueroa et al., 2010a*). Unsupervised clustering based on cytosine methylation demonstrated that pancreatic CAFs were epigenetically distinct from other non-cancer associated fibroblast controls that also included hepatic stellate cells. (*Figure 1A*). To determine the qualitative epigenetic differences between these groups we next performed a supervised analysis of the respective DNA methylation profiles. A volcano plot comparing the differences between mean methylation of individual loci between pancreatic CAFs and non-cancer associated fibroblasts demonstrated that pancreatic CAFs were characterized by widespread hypomethylation when compared to controls (5659

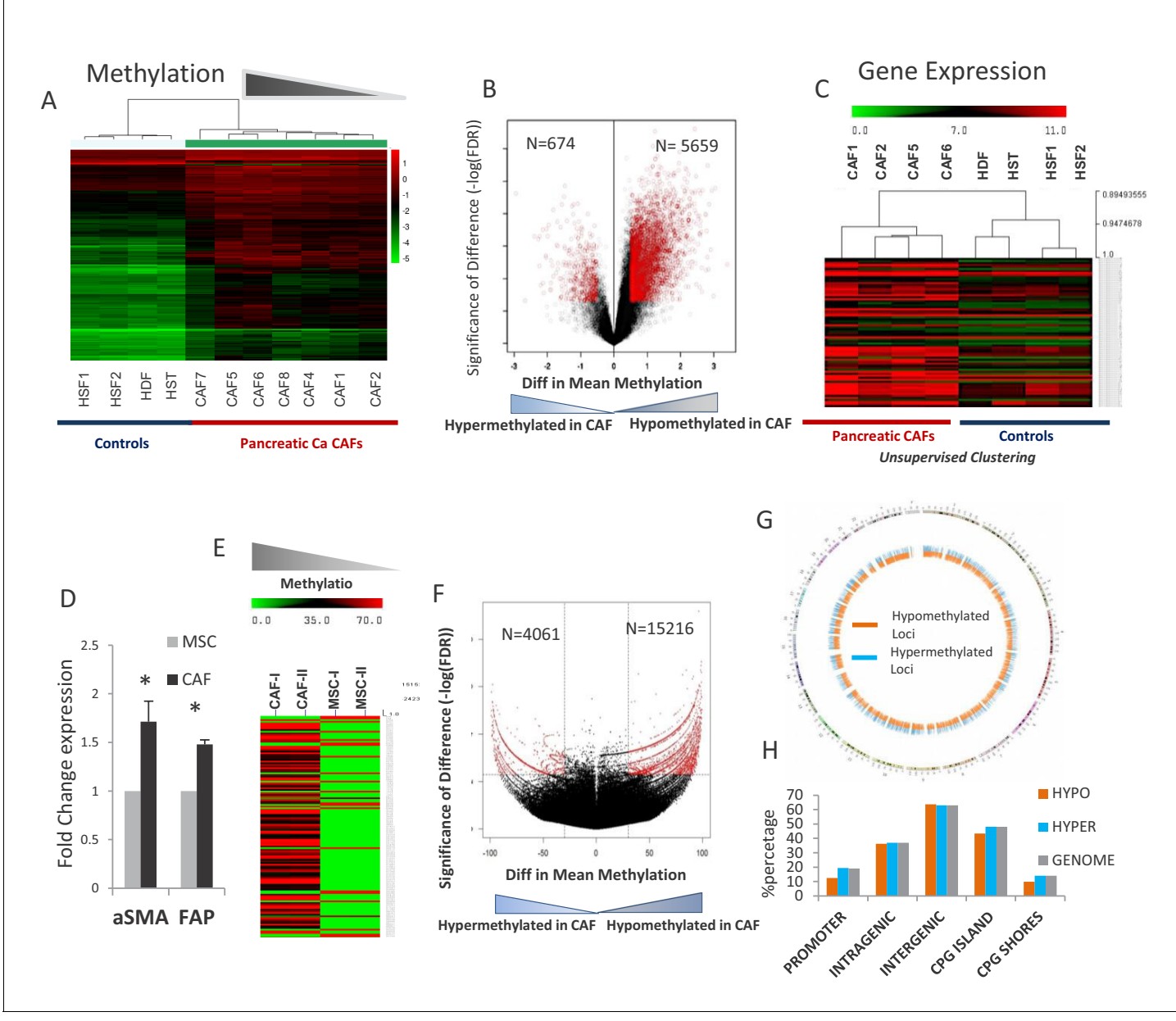

**Figure 1.** Widespread epigenetic and transcriptomic alterations are seen in pancreatic cancer associated fibroblasts. (**A**) Unsupervised clustering of cytosine methylation profiles from seven primary patient-derived pancreatic cancer associated fibroblasts (CAFs) and four healthy controls (Hst: Hepatic stellate cells, Hdf: Human dermal fibroblasts, Hsf: Human skin fibroblasts) shows that CAFs are epigenetically distinct (**B**) Volcano plot shows that majority of differentially methylated loci in primary patient-derived CAFs are hypomethylated when compared to controls (**C**) Unsupervised clustering of gene expression profiles shows transcriptomic differences between CAFs and controls (**D**) Three independent MSCs were exposed to PANC-1 conditioned media (CM) for 21 days and analyzed for α-SMA (*ACTA1*) and Fibroblast activation protein (FAP) expression. (T-test, p<0.05). (**E**) Unsupervised clustering based on cytosine methylation shows epigenomic differences between MSCs and de novo generated CAFs. Two independent experiments shown. (**F**) Volcano plot shows that the majority of differentially methylated loci in in vitro generated CAFs are hypomethylated (**G**) The differentially methylated regions in de novo generated CAFs are distributed throughout the genome as shown in the circos plot. H:Differentially methylated regions between MSC and de novo generated CAFs are present throughout the genome and mirror the distribution of HpaII sites in the genome. Hypo refers to Hypomethylated DMRs in CAFs. Hyper refers to Hypermethylated DMRs in CAFs when compared to MSCs. Genome refers to distribution of HpaII loci in the genome.

DOI: https://doi.org/10.7554/eLife.50663.002

demethylated *versus* 674 hypermethylated loci in CAFs) (*Figure 1B*). Gene expression analyses performed on a subset of CAFs also demonstrated transcriptomic differences when compared to controls (*Figure 1C*). To elucidate the genes that were epigenetically regulated, we analyzed the genes that were concurrently overexpressed and hypomethylated in pancreatic CAFs and observed that critical cellular pathways involved in cell survival, cell cycle and cell signaling were the most significantly deregulated by epigenetically altered genes (**Supp File 1**). Multiple genes that are known to be important for cell signaling, including secreted interleukins and chemokines such as IL1a, CCL5, CCL26, cellular receptors CXCR4, ICAM3 and signaling proteins MAPK3, MAPK7, JUN were among the easily recognizable genes that exhibited differential hypomethylation and were overexpressed in pancreatic CAFs. Since striking demethylation was observed in primary CAFs, we next wanted to validate these epigenetic changes at a higher resolution in an in vitro model. We generated CAFs from primary mesenchymal stem cells (MSCs) by exposing them to conditioned media from Panc-1 pancreatic cancer (PDAC-CM) cells for 21 days. This method has been shown to transform MSCs into CAFs that are functionally able to support the growth and invasion of malignant cells (*Mishra et al., 2008*) and resulted in cells with CAF like morphology and higher expression of bona fide CAF markers, aSMA (*ACTA1*) and FAP (*Figure 1D*). The methylome of MSCs and de novo CAFs was then studied using the HELP-tagging assay that uses massively parallel sequencing to generate genome wide CpG methylation profiles of >1.8 million loci (*Suzuki et al., 2010*). We observed that widespread cytosine demethylation was the dominant epigenetic change during transformation of MSCs into de novo CAFs (*Figure 1E,F*). Loss of methylation upon exposure to PDAC-CM was found to affect all parts of the genome (*Figure 1G,H*). Both hypo- and hypermethylated differentially methylated regions (DMRs) were distributed in various genomic locations in proportions that were comparable to the distribution of *Hpa*II sites (*Figure 1H*), thus demonstrating that epigenetic changes were occurring genome-wide during CAF transformation. This is in contrast to absence of genetic alterations in PDAC CAFs, as reported by *Walter et al. (2008)*.

## CXCR4 is hypomethylated and overexpressed in pancreatic cancer associated fibroblasts and supports neoplastic cell invasion

Integrative analysis between primary CAFs and de novo CAFs showed a common set of 130 unique promoters that were aberrantly methylated (120 hypomethylated and 10 hypermethylated) (*Figure 2A*, Venn diagram). This conserved epigenetic signature of CAFs (*Supplementary file 2*) was able to clearly separate these cells from normal controls and MSCs in supervised clustering (*Figure 2A*, Bottom panel). The gene promoter encoding for the CXCR4 receptor was found to be hypomethylated in both primary and de novo generated CAFs, and significantly demethylated CpGs were present in CpG 'shores', that have been shown to be sites of differential methylation in cancer (*Irizarry et al., 2009*) (*Figure 2B,C*). Although recent empirical studies have found upregulation of CXCR4 protein in PDAC cells by immunohistochemical assessment (*Bachet et al., 2012*), the role of this receptor on the stromal cells have not been studied.

To determine the functional role of CXCR4 expression on pancreatic CAFs, we used specific siRNAs against CXCR4 that were able to significantly decrease *CXCR4* expression in MSC-derived de novo CAFs (*Figure 2D*, *Figure 2—figure supplement 1*). Matrigel transwell double chamber invasion assays with PDAC (PANC-1) cells revealed increased invasion of the neoplastic cells in the presence of de novo generated CAF cells (*Figure 2E*). The increased invasiveness of PDAC cells on co-culture was abrogated with RNAi-mediated knockdown of *CXCR4* in the CAFs (t-test, *P* Value < 0.05) (*Figure 2E,F*, *Figure 2—figure supplement 1*). A specific inhibitor of CXCR4, AMD-3100, also led to decreased invasion of PANC-1 cells when cultured with de novo CAFs. (*Figure 2G*). These were validated in another highly metastatic Pa03C PDAC cell line (*Figure 2—figure supplement 1*), supporting a role for stromal CXCR4 blockade in attenuating tumor progression within the neoplastic compartment. Finally, gene expression profiling in two sets of CAFs after *CXCR4* knockdown revealed significant alterations in the transcriptional profile (*Supplementary file 3*), including reduced expression of IL8 and other secreted ligands that are known to be tumor microenvironment regulators (*Matsuo et al., 2009*; *Fang et al., 2016*) (*Figure 2H*).

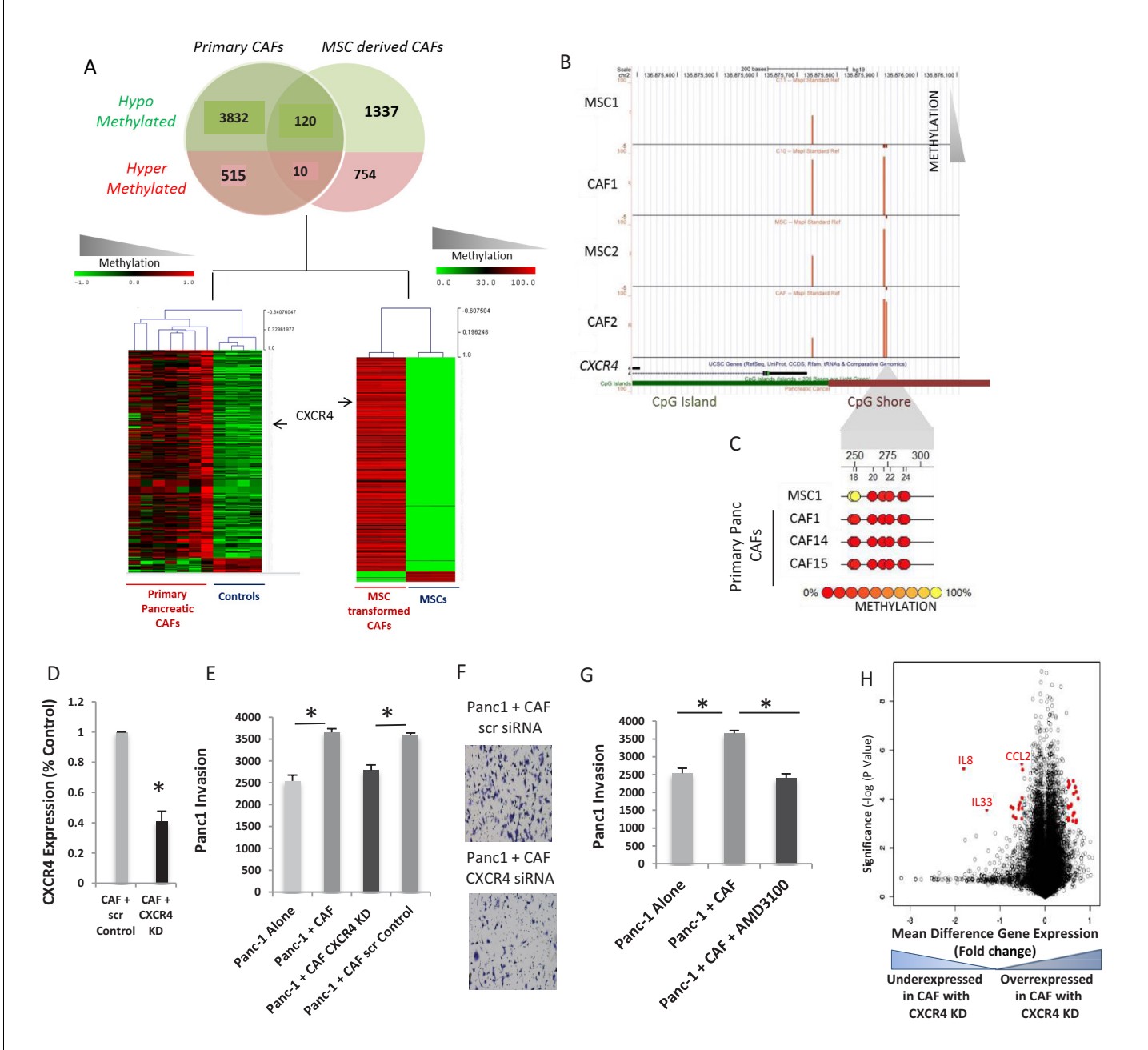

**Figure 2.** CXCR4 is demethylated and overexpresed in CAFs and increases pancreatic cancer invasiveness. (**A**) A set of 120 transcripts are commonly hypomethylated in primary patient-derived CAFs and de novo generated CAFs and includes *CXCR4*. (**B,C**) The *CXCR4* promoter is demethylated in primary patient-derived CAFs as seen by the HELP assay (**B**) and quantitative MassArray Epityper analysis (**C**). (**D - F**) CXCR4 knockdown in de novo CAFs leads to abrogation of the increased invasion of Panc1 cells on co-culture. (N = 3, p value<0.05) (**G**) Co-culture with de novo CAFs leads to increased transwell invasion by Panc-1 cells, that is abrogated after treatment of CAFs with CXCR4 inhibitor AMD-3100 (N = 3, p value<0.05) H: Gene expression profiling of CAFs with CXCR4 knockdown reveals signficantly downregulated (*left panel*) and upregulated (*right panel*) transcripts. Salient examples of downregulated transcripts include interleukins IL-8 and IL-33 and the chemokine CCL2.

DOI: https://doi.org/10.7554/eLife.50663.003

The following figure supplement is available for figure 2:

**Figure supplement 1.** Inhibition of CXCR4 reduces PDAC cell invasion in de novo generated CAFs.
DOI: https://doi.org/10.7554/eLife.50663.004

## Tumor mediated lactate flux leads to production of alpha ketoglutarate, TET activation and increased cytosine hydroxymethylation in stromal cells

Next, we wanted to determine whether a diffusible factor secreted by PDAC cells could facilitate the epigenomic reprogramming and demethylation of MSCs to CAFs. Lactate is produced by PDAC cells via lactate dehydrogenase (LDH) enzyme during glycolysis (*Le et al., 2010*) and has been shown to be an important mediator of metabolic pathways that can regulate the demethylase TET enzymes (*Intlekofer et al., 2015*; *Figueroa et al., 2010b*). Thus, we wanted to evaluate whether paracrine lactate, secreted by PDAC cells, could be incorporated by MSCs and result in observed epigenetic changes through modulation of TET activity. Metabolomics analysis using uniformly [13]C-labeled lactate in the media as a tracer revealed that primary human MSCs can uptake lactate and convert it to pyruvate and various Krebs cycle intermediates including citrate, alpha-keto glutarate (aKG), fumarate, malate, and aspartate (*Figure 3A*, *Figure 3—figure supplement 1* showing metabolities in parallel experiments using low glucose conditions). We had previously observed a similar metabolic crosstalk between ovarian cancer cells and CAFs, which utilized cancer-secreted lactate as a carbon source (*Yang et al., 2016*).

Since aKG is an important cofactor for TET enzymes, we next determined whether lactate secreted by PDAC-conditioned media could lead to TET activation and increased hydroxymethylcytosine (5hmC) levels in stromal cells. MSCs were exposed to conditioned media from PANC-1 cells treated with an LDH inhibitor or control media for 14 days to induce generation of de novo CAFs. Control PANC-1 conditioned media per se led to significant TET activation (*Figure 3B*) and increase in 5hmC levels in resulting de novo CAFs (*Figure 3C*). These effects were, however, not observed when MSCs were exposed to conditioned media from LDH inhibitor-treated PDAC cells. Conditioned media from PDAC cells with LDH knockdown with siRNAs was also able to abrogate the increase in expression of CAF markers (aSMA (*ACTA1*), FSP and Vimentin) during CAF conversion (*Figure 3—figure supplement 2A*). Furthermore, exposure of MSCs to exogenous lactate in media was also able to increase aKG levels, TET activity and 5hmC levels after 2 weeks of exposure (*Figure 3D,E*, *Figure 3—figure supplement 2*), demonstrating the role of this exogenous metabolite in epigenetic reprogramming of MSCs to de novo CAFs (Proposed Model in *Figure 3E*). TET activity increased in a dose dependant manner with exogenous lactate and was abrogated by inhibition of mitochondrial pyruvate carrier inhibitor UK5099 (*Figure 3—figure supplement 2*). Since 2hydroxyglutarate (2HG) and fumarate are inhibitors of TET enzymes, we also observed that exogenous lactate was able to reduce 2HG/aKG and fumarate/aKG ratios in metabolic flux experiments (*Figure 3—figure supplement 1*). Additionally, exogenous cell permeable aKG was also able to increase TET activity in CAFs (*Figure 3—figure supplement 2*).

## 5hmc gains are seen during MSC to CAF conversion

To determine the genes that acquire 5hmC during CAF conversion at a high resolution, we generated CAFs from MSCs exposed to Panc1 conditioned media and used the cells for genome wide 5hmC analysis using Oxidative bisulfite sequencing (OXBS). The 5hmC gains were found to occur throughout the genome (*Figure 4A*). Genes affected by 5hmC gains were found to group into important regulatory pathways (*Figure 4B*). Gene associated with 5hmC gains included CAF markers αSMA (*ACTA1*), FSP1 (*S100A4*), and Collagen (COL3A1, associated with fibrotic reaction seen in pancreatic cancer) (*Figure 4C–E*). We also observed 5hmC gain at the CXCR4 promoter as well as downstream enhancer (*Figure 4F*). The important TGF-β mediator, SMAD2 also acquired 5hmC (*Figure 4G*) and a transcription factor motif analysis of all sites of 5hmC gain revealed enrichment for smad2 and smad3 binding motifs (*Table 1*). TGF-beta mediated smad activation is an important regulator of fibrosis and has been shown to be activated in CAFs (*Biffi et al., 2019*).

## Increased stromal 5hmc and *CXCR4* expression is seen in in primary human PDAC and murine KPC PDAC tumors

We next wanted to determine the magnitude of increased 5hmC and CXCR4 in stromal cells in large cohort of human primary PDAC samples. 5hmC (*Figure 5A*) and CXCR4 (*Figure 5B*) immunohistochemical staining was done on human PDAC TMAs and grading of intensity of stain in the tumor stromal CAFs was estimated. We observed that most CAF like cells in PDAC samples were positive

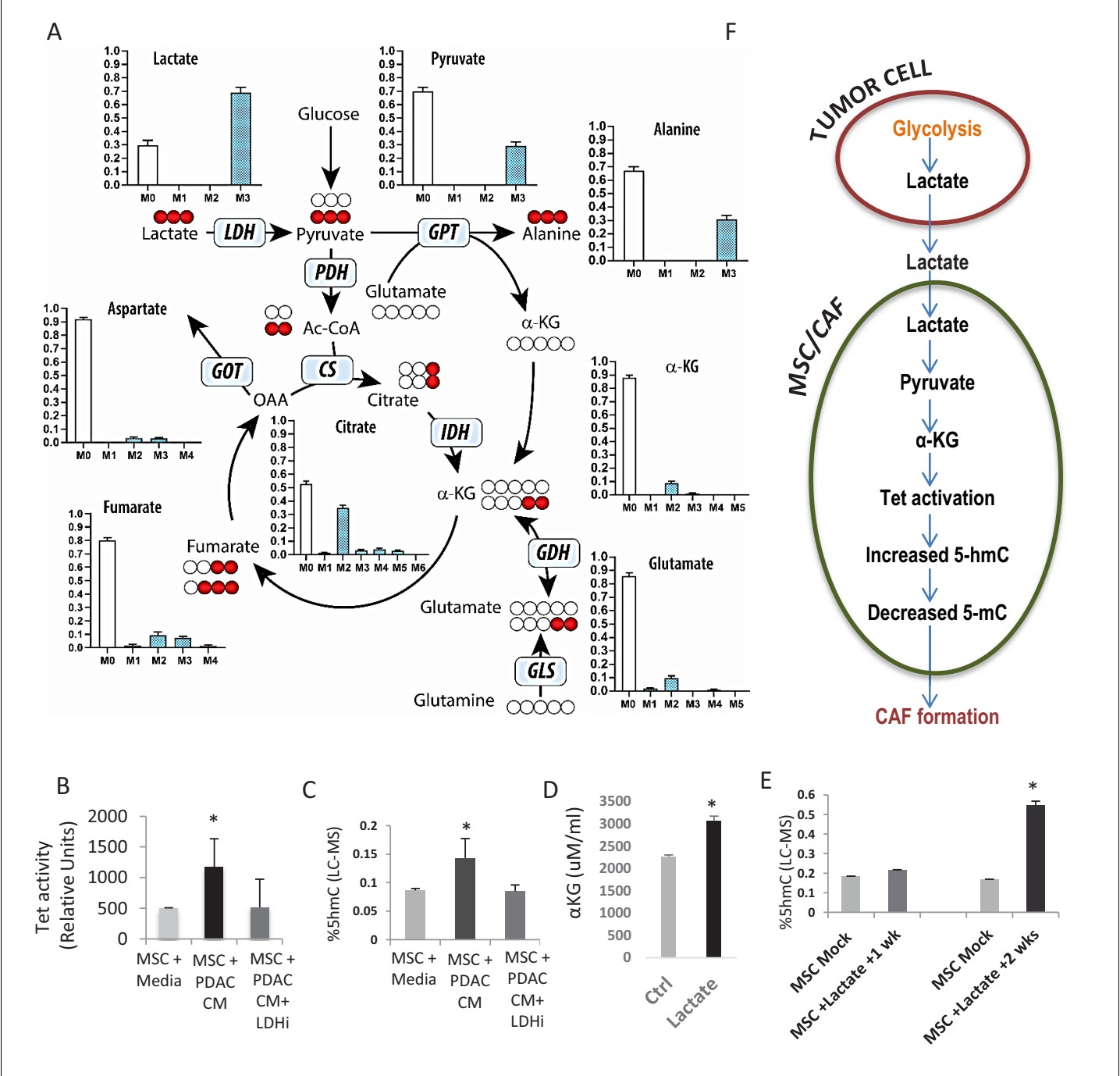

**Figure 3.** Increased 5hmC is seen in de novo CAFs generated from MSCs and is regulated by a paracrine lactate mediated metabolic flux. (**A**) Mapping of carbon atoms transition using U-$^{13}C_3$-lactate. Mass isotopomer distribution (MID) shows uptake of labelled lactate by MSCs cultured in α-MEM leading to the generation of α-KG (M2 αKG) via the Krebs cycle. (**B**) Conditioned media (CM) from mock treated and LDH inhibitor (FX11) treated Panc-1 (PDAC) cells was collected. MSCs were exposed to media alone, control conditioned media (PDAC CM), or LDH inhibitor treated conditioned media for 14 days in order to generate de novo CAFs (dn-CAF). TET enzymatic activity increases in dn-CAFs after exposure to PDAC CM and is abrogated after exposure to CM from LDH inhibitor treated Panc-1 cells (N = 2, p<0.05). (**C**) Quantitative analysis of 5-hMC levels by LC-MS demonstrates significant increase within dn-CAFs after treatment with Panc-1 CM, that is abrogated after exposure to CM from LDH inhibitor treated Panc-1 cells (N = 2, p<0.05). (**D**) CAFs were exposed to exogenous lactate and α-KG levels were analysed by ELISA. (N = 2, p<0.05) (**E**) 5hmC analysed in the resulting dn-CAFs by LC-MS. After a 2 week exposure to exogenous lactate, significantly increased 5hmC is observed in the dn-CAFs (N = 2, p<0.05). (**F**) Schematic model of lactate flux from tumor cells to MSCs during CAF differentiation, leading to aKG generation, TET activation and conversion of 5mC into 5hmC.

*Figure 3 continued on next page*

*Figure 3 continued*

DOI: https://doi.org/10.7554/eLife.50663.005

The following figure supplements are available for figure 3:

**Figure supplement 1.** Metabolite tracking in CAF cells.

DOI: https://doi.org/10.7554/eLife.50663.006

**Figure supplement 2.** Lactate increases TET activity in CAFs.

DOI: https://doi.org/10.7554/eLife.50663.007

for 5hmC and CXCR4 (1+ to 3+ staining intensity), (*Figure 5C*). Total PDAC samples examined for 5hmC were 254 and for CXCR4 were 261.

Next, we wanted to evaluate whether increased stromal 5hmC and CXCR4 was observed in mouse model of PDAC also. PDAC samples from KPC (*Kras* mutant, *tp53* mutant) (*Olive et al., 2009*) mouse model of were obtained and immunostained for 5hmC and CXCR4. CAF like stromal cells in PDAC tumors from KPC mice were found to be positive for both 5hmC and CXCR4 staining in all tumors examined (*Figure 5D,E*).

To validate at single cell levels, we next analyzed single cell RNA-seq (scRNAseq) data from samples obtained from precancerous low grade intraductal papillary mucinous neoplasm (LG IPMN), high grade intraductal papillary mucinous neoplasm (HG IPMN) and frank pancreatic ductal adenocarcinoma (PDAC) (*Bernard et al., 2019*). Stromal cell populations positive for alphaSMA (*ACTA1*) (*Figure 5G*) and fibroblast activated protein (*Figure 5H*) were found to cluster distinctly from malignant cells in tSNE plots. Most of stromal cells were seen in high grade IPMN and PDAC samples (*Figure 5I*). *CXCR4* expression was seen in 14/181 (8%) stromal cells and correlated with cells with higher collagen expression, that is seen in activated CAF phenotypes (*Öhlund et al., 2017*) (TTEst, P Val = 0.02)(*Figure 5K*).

## TET-deficient MSCs lead to inefficient CAF conversion and reduced tumor growth in vivo

Having demonstrated that exposure to PDAC-conditioned media lead to TET activation with a concomitant increase in 5hmC levels within CAFs, we next wanted to determine the functional role of TET enzymes during MSC to de novo CAF differentiation. MSCs were obtained from TET2 KO mice (*Dawlaty et al., 2014*) and controls and were co-cultured with conditioned media from murine PDAC cells derived from Kras, p53 mutant tumors ('KPC' cells) (*Torres et al., 2013*). WT MSCs acquired fibroblastic 'CAF-like' appearance after exposure to PDAC conditioned media, while TET2 KO MSCs generally retained their original morphology (*Figure 6A*). Additionally, TET2 KO MSCs that were exposed to CM led to significantly less KPC PDAC cell invasion in matrigel when compared to WT controls (*Figure 6B,C*). Next, to determine the functional role of TET demethylase in CAF generation in vivo, we co-injected murine KPC cells with MSCs from TET2 KO mice and controls into immune-deficient mice. Co-injected murine PDAC cells with TET2KO MSCs had significantly slower growth rates in vivo when compared to controls (*Figure 6D*). Explanted allografts with TET2 KO MSCs were significantly smaller (*Figure 6E,F*) and histologically revealed significantly less cells with a CAF phenotype, on staining with aSMA (*ACTA1*) (*Figure 6G,H*), thus demonstrating less efficient CAF conversion and tumor supporting capabilities in vivo.

## Discussion

The tumor microenvironment plays a critical role in promoting the growth and invasion of cancer cells (*Karnoub et al., 2007*; *Orimo and Weinberg, 2006*). Cancer cells can recruit MSCs from the marrow and facilitate their transformation into activated CAFs, through a plethora of paracrine signals, such as chemokines (*Mishra et al., 2011*; *Quante et al., 2011*). One aspect of the cancer cell 'secretome' within the immediate juxtatumoral milieu that has not been fully examined pertains to the role of secreted metabolites, or metabolic intermediates. We demonstrate that conversion of MSCs into CAFs is associated with widespread epigenomic reprogramming. Specifically, we establish that tumor generated lactate can be incorporated by stromal cells and can potentially induce epigenomic changes via increased production of aKG. Notably, aKG is an essential cofactor for the

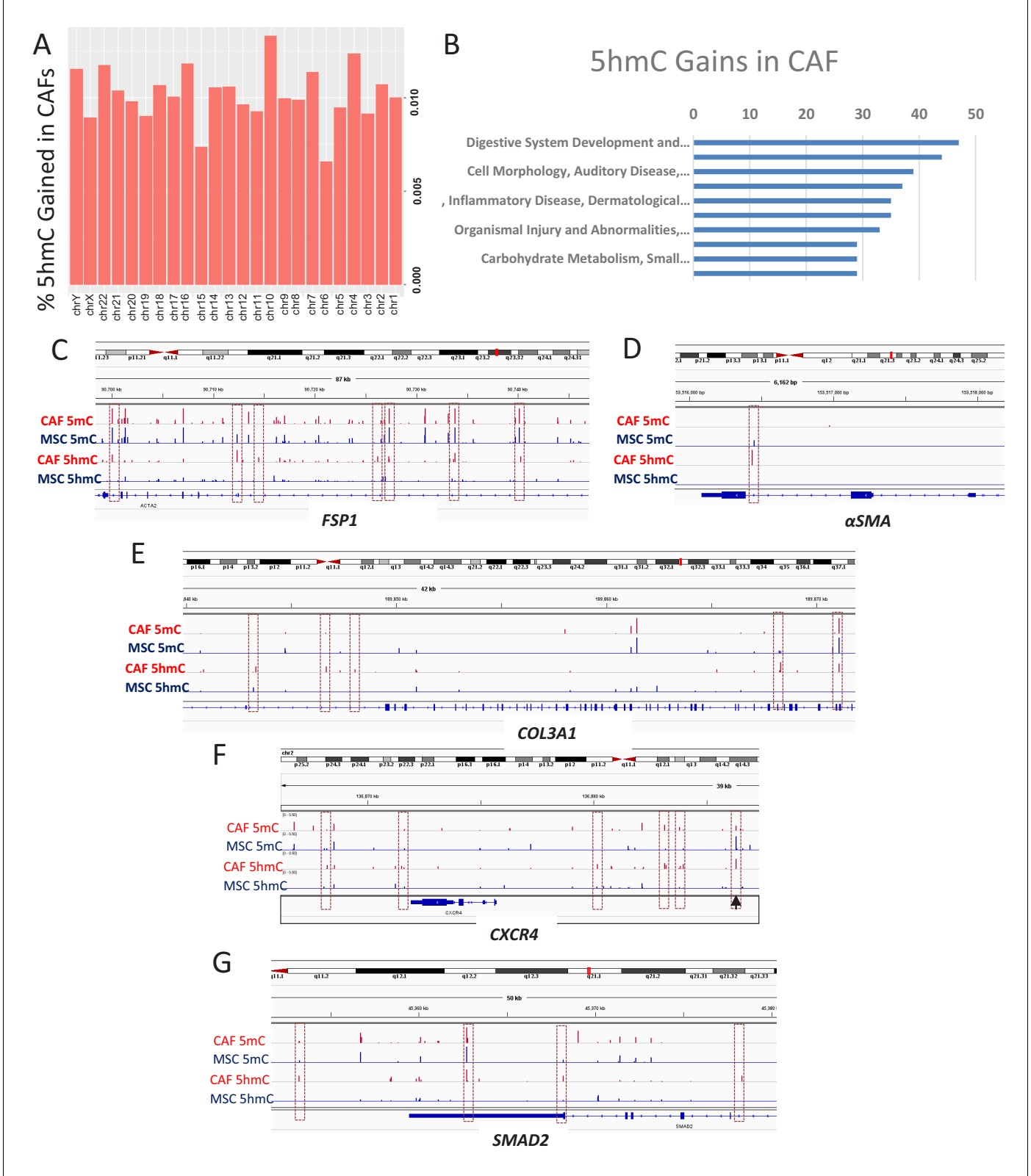

**Figure 4.** 5hmC gains are seen during MSC to CAF conversion and occur at gene associated with cancer associated fibroblasts. (**A**) Genome wide 5hmC analysis was done by OXBS in MSCs and converted CAFs. 5hmC gains are depicted as percentage gained (CAF/MSC) and were seen to occur throughout the genome (**B**) 5hmC gains in CAFs were seen to occur at important genetic pathways C-G: 5hmC gains occurred with corresponding decrease in 5mC at genes related to CAF conversion (FSP1 (*S100A4*), aSMA (*ACTA1*)); Collagen production (COL3A1), CXCR4, and the SMAD2.

DOI: https://doi.org/10.7554/eLife.50663.008

**Table 1.** Transcription factor sites that are enriched around 5hmC gains during CAF conversion.

| Rank | Name | P-value | q-value (Benjamini) | # Target Sequences with Motif | % of Targets Sequences with Motif | # Background Sequences with Motif | % of Background Sequences with Motif | Motif |
|---|---|---|---|---|---|---|---|---|
| 1 | Atf3(bZIP)/GBM-ATF3-ChIP-Seq(GSE33912)/Homer | 1.00E-07 | 0 | 344 | 7.86% | 2593.1 | 5.81% | |
| 2 | AP-1(bZIP)/ThioMac-PU.1-ChIP-Seq(GSE21512)/Homer | 1.00E-06 | 0 | 383 | 8.75% | 2995.7 | 6.71% | |
| 3 | Jun-AP1(bZIP)/K562-cJun-ChIP-Seq(GSE31477)/Homer | 1.00E-05 | 0.0003 | 112 | 2.56% | 722.1 | 1.62% | |
| 4 | BMAL1(bHLH)/Liver-Bmal1-ChIP-Seq(GSE39860)/Homer | 1.00E-05 | 0.0003 | 857 | 19.57% | 7567 | 16.95% | |
| 5 | Fosl2(bZIP)/3T3L1-Fosl2-ChIP-Seq(GSE56872)/Homer | 1.00E-05 | 0.0003 | 149 | 3.40% | 1040.8 | 2.33% | |
| 6 | BATF(bZIP)/Th17-BATF-ChIP-Seq(GSE39756)/Homer | 1.00E-05 | 0.0003 | 322 | 7.35% | 2570.8 | 5.76% | |
| 7 | Fra1(bZIP)/BT549-Fra1-ChIP-Seq(GSE46166)/Homer | 1.00E-04 | 0.0006 | 266 | 6.07% | 2089.9 | 4.68% | |
| 8 | Smad2(MAD)/ES-SMAD2-ChIP-Seq(GSE29422)/Homer | 1.00E-04 | 0.0011 | 652 | 14.89% | 5725.2 | 12.83% | |
| 9 | NPAS2(bHLH)/Liver-NPAS2-ChIP-Seq(GSE39860)/Homer | 1.00E-04 | 0.0026 | 543 | 12.40% | 4734.1 | 10.61% | |
| 10 | Smad4(MAD)/ESC-SMAD4-ChIP-Seq(GSE29422)/Homer | 1.00E-03 | 0.003 | 655 | 14.96% | 5819.3 | 13.04% | |
| 11 | Smad3(MAD)/NPC-Smad3-ChIP-Seq(GSE36673)/Homer | 1.00E-03 | 0.0035 | 1238 | 28.27% | 11533 | 25.84% | |
| 12 | Usf2(bHLH)/C2C12-Usf2-ChIP-Seq(GSE36030)/Homer | 1.00E-03 | 0.0042 | 200 | 4.57% | 1574.5 | 3.53% | |
| 13 | HIF2a(bHLH)/785_O-HIF2a-ChIP-Seq(GSE34871)/Homer | 1.00E-03 | 0.0066 | 178 | 4.06% | 1393.5 | 3.12% | |
| 14 | CEBP:CEBP(bZIP)/MEF-Chop-ChIP-Seq(GSE35681)/Homer | 1.00E-03 | 0.0141 | 107 | 2.44% | 789.3 | 1.77% | |
| 15 | MafA(bZIP)/Islet-MafA-ChIP-Seq(GSE30298)/Homer | 1.00E-02 | 0.023 | 315 | 7.19% | 2709.1 | 6.07% | |
| 16 | Bach2(bZIP)/OCILy7-Bach2-ChIP-Seq(GSE44420)/Homer | 1.00E-02 | 0.023 | 86 | 1.96% | 622.7 | 1.40% | |
| 17 | Brachyury(T-box)/Mesoendoderm-Brachyury-ChIP-exo(GSE54963)/Homer | 1.00E-02 | 0.0326 | 134 | 3.06% | 1056.7 | 2.37% | |
| 18 | USF1(bHLH)/GM12878-Usf1-ChIP-Seq(GSE32465)/Homer | 1.00E-02 | 0.0387 | 190 | 4.34% | 1574.4 | 3.53% | |

*Table 1 continued on next page*

Table 1 continued

| Rank | Name | P-value | q-value (Benjamini) | # Target Sequences with Motif | % of Targets Sequences with Motif | # Background Sequences with Motif | % of Background Sequences with Motif | Motif |
|---|---|---|---|---|---|---|---|---|
| 19 | n-Myc(bHLH)/mES-nMyc-ChIP-Seq(GSE11431)/Homer | 1.00E-02 | 0.0511 | 272 | 6.21% | 2354.1 | 5.27% | |
| 20 | Fli1(ETS)/CD8-FLI-ChIP-Seq(GSE20898)/Homer | 1.00E-02 | 0.0528 | 439 | 10.03% | 3953.7 | 8.86% | |

DOI: https://doi.org/10.7554/eLife.50663.009

functionality of TET demethylates (*Carey et al., 2015*). Studies in embryonic stem cells demonstrated the ability of lactate in reprogramming stem cells via epigenetic alterations (*Carey et al., 2015*). It has also been shown that tumor lactate can cause pleiotropic effects in surrounding immune cells, as well as in the tumor cells themselves (*Matilainen et al., 2017*). Our data demonstrate that this diffusible factor can be a potential critical mediator that facilitates CAF differentiation in the vicinity of glycolytic proliferative tumors such as PDAC. Lactate inhibitors are being developed for anti-tumor activity (*Le et al., 2010*; *Rajeshkumar et al., 2015*) and our data suggests that these inhibitors may act via effects on tumor microenvironment also.

Epigenetic reprogramming, which manifests as widespread loss of DNA methylation and gain of cytosine hydroxymethylation at selective promoters, is seen in both MSC-derived and primary (patient-derived) pancreatic CAFs. Increased lactic acid can result in acidic environment and it is possible that the change in pH can also influence epigenetic states. Previous studies have shown that a low acidic ph leads to increases in 2HG more than aKG (*Nadtochiy et al., 2016*). 2HG increase generally leads to decreased Tet activity. Thus in our model, a low ph (due to increase lactic acid) should not account for the high Tet activity and consequent decreased 5mC that we see; suggesting that lactic acid mediated increases in aKG and Tet activity were not influenced by changes in local pH. Loss of DNA methylation has been described mainly during developmental processes of early embryo development, and also during differentiation of hematopoietic stem cells to committed red cell progenitors (*Shearstone et al., 2011*; *Yu et al., 2013*). In fact, involvement of a large proportion of the genome by demethylation is rarely outside of developmental processes. Also, though hypomethylation has been shown in some solid tumors (*Timp and Feinberg, 2013*; *Alvarez et al., 2011*), it has not been studied at single base pair resolution in the tumor microenvironment. Epigenetic studies in the tumor microenvironment have mainly been single locus studies that have focused on hypermethylation of specific gene promoters during CAF transformation. Our findings show that widespread demethylation occurs during in vitro transformation of MSCs to CAFs, and is recapitulated in primary patient-derived CAF samples. In fact, genome wide analysis of 5hmC showed that 5hmC acquisition was seen in genes that have been associated with pancreatic cancer associated CAFS. 5hmC is an epigenetic modification that is obtained from oxidation of 5mC marks and is an intermediary step towards demethylation (*Ko et al., 2010*). It is postulated that 5hmC can act as an independent regulatory activating mark and is associated with sites of active transcription (*Bhattacharyya et al., 2017*; *Madzo et al., 2014*). Furthermore, our findings are consistent with a study in gastric cancer that also observed loss of methylation and was consistent with our findings. (*Jiang et al., 2008*) A recent immunohistochemical study in a murine model of PDAC also observed loss of methylation in the microenvironment (*Shakya et al., 2013*) though it did not study locus specific changes.

Our data shows that the chemokine receptor, CXCR4, is upregulated in CAFs and is associated with loss of promoter methylation. Interestingly the promoter CpGs that are demethylated were not located in CpG islands, but in the neighboring CpG shore. CpG shores flank CpG islands and have been shown to areas where differential methylation can occur in cancer (*Irizarry et al., 2009*). CXCR4 is a well-studied receptor in stem cells and cancer models (*Rettig et al., 2012*; *Wong and Korz, 2008*) and most studies have evaluated its expression on the tumor cells. We show that in addition to its roles on tumor cells, its expression on stromal cells is also functionally important in

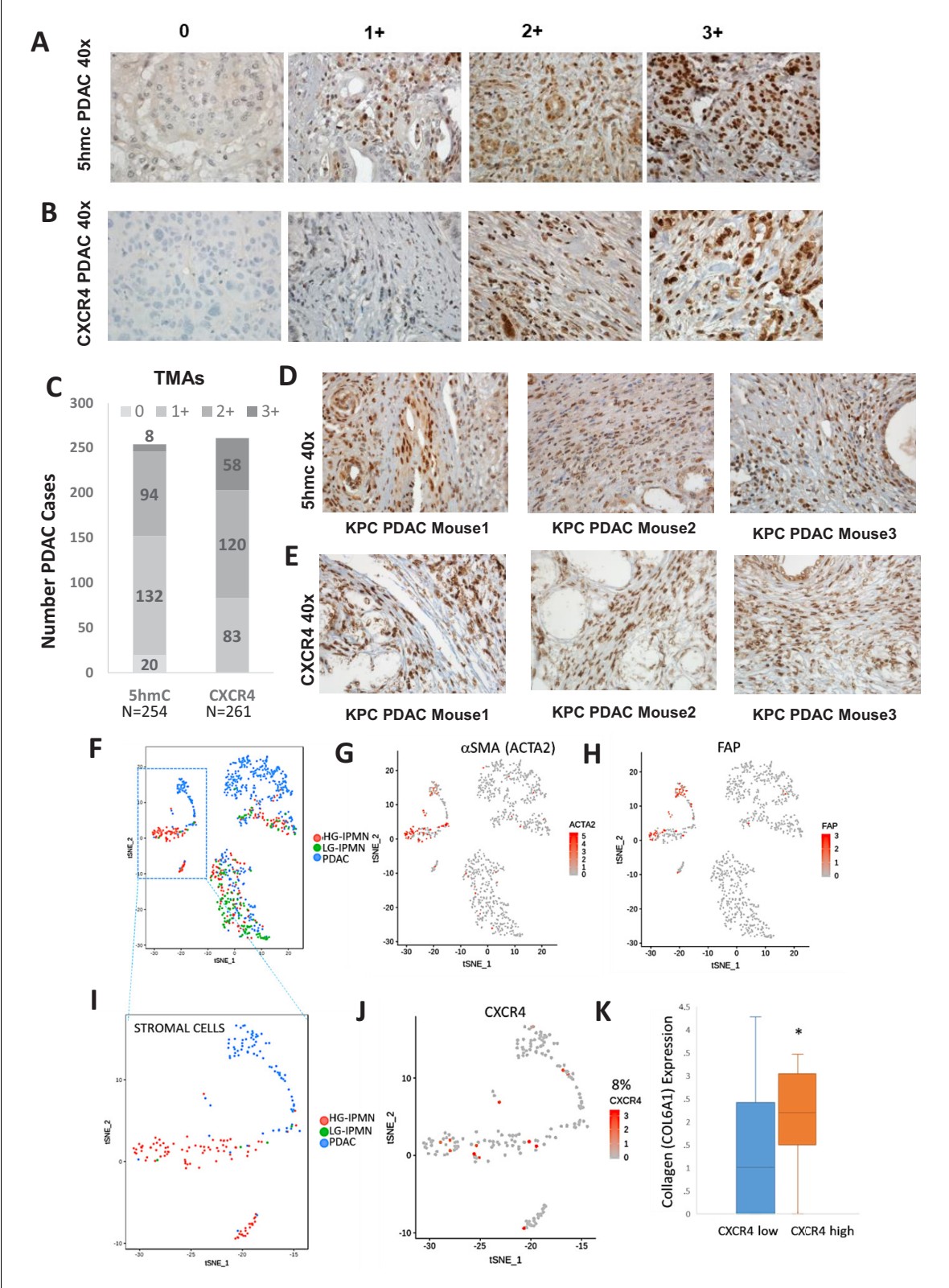

**Figure 5.** Increased 5hmC and CXCR4 expression is seen in primary human PDAC and murine KPC PDAC tumors. (**A, B**) 5hmC (**A**) and CXCR4 (**B**) immunohistochemical staining was done on human PDAC TMAs and grading of intensity of stain in the tumor stromal CAFs was estimated. (**C**) Most CAF like cells in PDAC samples were positive for 5hmC and CXCR4 (1+ to 3+ intensity). Total PDAC samples examined for 5hmC were 254 and for CXCR4 were 261. (**D,E**) 5hmC (**D**) and CXCR4 (**E**) immunohistochemical staining was done on mouse PDAC tumors obtained from 3 KPC mice. CAF like

*Figure 5 continued on next page*

*Figure 5 continued*

stromal cells were positive for 5hmC and CXCR4. (**F**) Single cell RNA-seq from low grade intraductal papillary mucinous neoplasm (LG IPMN), high grade intraductal papillary mucinous neoplasm (HG IPMN) and frank pancreatic ductal adenocarcinoma (PDAC) was conducted and shown in tSne plot. (**G,H**) Stromal cell populations cluster distinctly and are positive for α-SMA (*ACTA1*) and fibroblast activated protein. (**I**) Most of the stromal cells are seen in High grade IPMN and PDAC samples J,K: CXCR4 expression is seen in 14/181 (8%) stromal cells and correlates with cells with higher collagen expression, seen in activated CAF phenotypes (T-Test, P Vccal = 0.02).

DOI: https://doi.org/10.7554/eLife.50663.010

regulating tumor cell invasion. Since host stromal cells and the cancer cells cross-talk via a large variety of soluble factors, chemokines are important mediators of these processes. It has been shown that CAFs (*Mishra et al., 2008*) as well as tumor cells (*Quante et al., 2011*) secrete SDF-1/CXCL12,

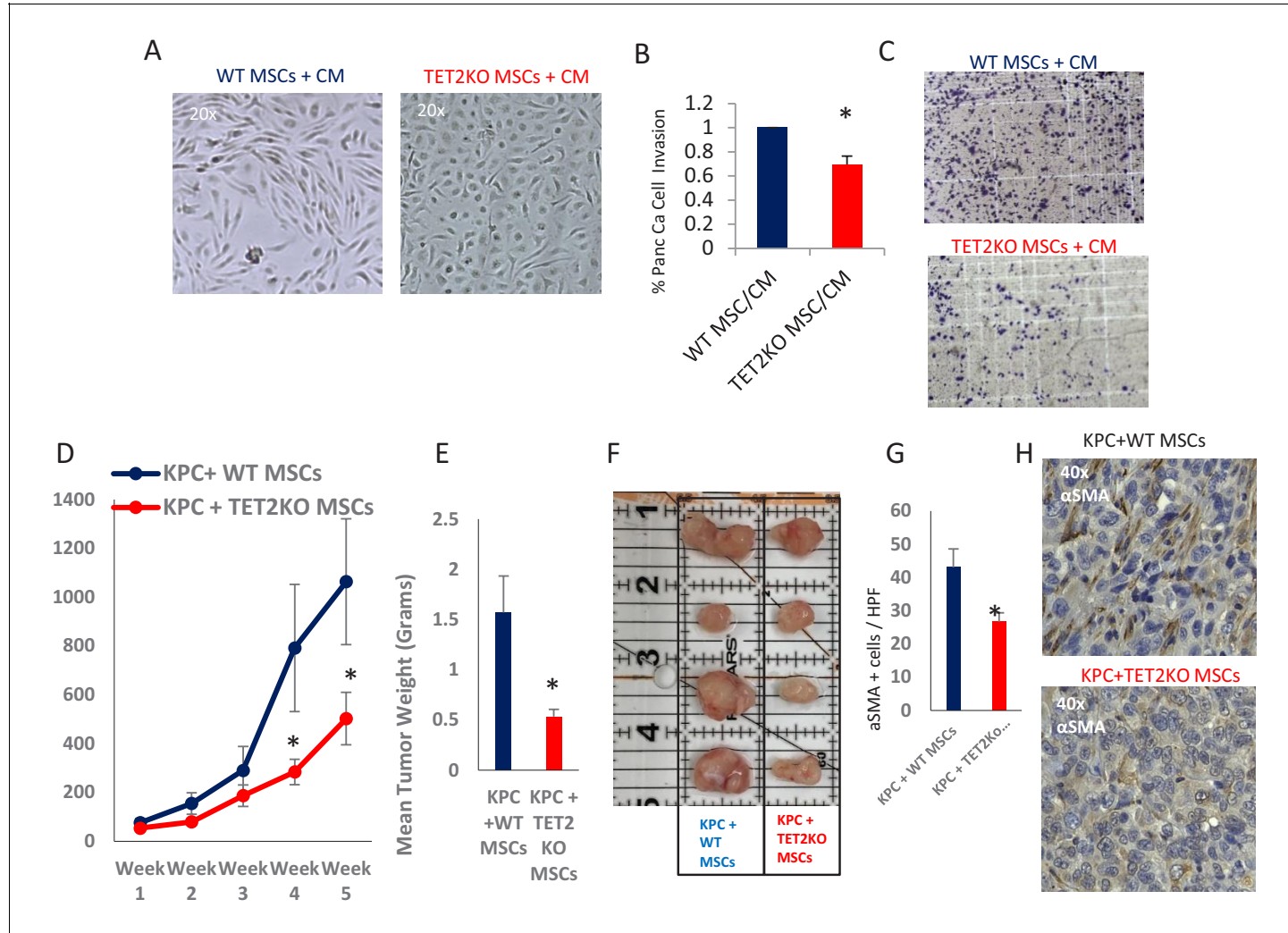

**Figure 6.** TET2 inactivation in MSCs leads to reduced tumor growth and inefficient CAF conversion in vivo. (**A**) Wildtype and TET2 KO MSCs were exposed to conditioned media from KPC PDAC cells for 14 days. TET2 KO MSCs demonstrated decreased fibroblastic appearance when compared to wildtype controls. (**B,C**) MSCs from TET2 KO treated with conditioned media from KPC PDAC cells led to significantly less KPC invasion in matrigel assays when compared with wildtype controls (N = 3 independent experiments, p value<0.05) (**D**) MSCs from TET2 KO mice and WT controls were co-injected subcutaneously with KPC PDAC cells in NSG mice. Tumor growth was significantly reduced in mice injected with TET2 KO stromal cells (N = 9 (TET2 KO/KPC) and N = 7 (WT/KPC), *P* Value < 0.05) (**E,F**) Tumor sizes (**B**) and weights (**C**) are significantly reduced from KPC allografts co-injected with TET2KO MSCs (*P* Value < 0.05) G,H: Number of aSMA (*ACTA1*) positive cancer associated fibroblasts are reduced in KPC allografts co-injected with TET2KO stromal cells when compared to KPC allografts co-injected with control MSCs (N = 4 in each group, p value<0.01).

DOI: https://doi.org/10.7554/eLife.50663.011

which is the ligand for CXCR4. A recent study demonstrated that overexpression of the CXCR4 ligand, SDF-1 in gastric cells can induce myofibroblast expansion and is consistent with the role of CXCR4 in pancreatic cancer (*Quante et al., 2011*; *Shibata et al., 2013*). The specific inhibitor of CXCR4, AMD-3100, is a FDA approved drug for hematopoietic stem cell mobilization. Our results demonstrate a role of this drug in inhibition of tumor stromal crosstalk. We also observed that production of secretory factors, such as interleukin-8 (IL8) by CAFs, is reduced upon CXCR4 knockdown. IL8 is a well-known regulator of cell motility and has been shown to be a regulator of pancreatic cancer cell invasion and growth (*Matsuo et al., 2009*; *Delitto et al., 2017*; *Sparmann and Bar-Sagi, 2004*).

Numerous studies have shown that depletion of stroma in PDAC can lead to enhanced efficacy of chemotherapy (*Feig et al., 2012*; *Öhlund et al., 2017*; *von Ahrens et al., 2017*; *Ireland et al., 2016*). Conversely, a murine study demonstrated that depletion of CAFs can accelerate PDAC metastases (*Özdemir et al., 2014*). As the prognosis of PDCA remains dismal, various clinical approaches are being attempted to target CAFs, and will further clarify the role of these cells in carcinogenesis. Our studies support the use of agents that disrupt the cross talk between malignant cells and the stroma and suggest that CXCR4 inhibitors may have a potential therapeutic role in pancreatic cancer that should be tested in future studies.

## Materials and methods

### DNA methylation analysis by HELP assay

Genomic DNA was isolated from primary CAF samples and controls with the use of a standard high-salt procedure, and the HELP assay was carried out as previously described (*Alvarez et al., 2011*; *Khulan et al., 2006*). The assay uses comparative isoschizomer profiling, interrogating cytosine methylation status on a genomic scale. Briefly, genomic DNA from the samples was digested by a methylcytosine-sensitive enzyme *Hpa*II in parallel with *Msp*I, which is resistant to DNA methylation, and then the *Hpa*II and *Msp*I products were amplified by ligation-mediated PCR. Both amplified fractions were submitted to Roche-NimbleGen, Inc (Madison, WI) for labeling and hybridization onto a human hg17 custom-designed oligonucleotide array (50-mers) covering 25,626 HpaII amplifiable fragments (HAF) located at gene promoters. HpaII amplifiable fragments are defined as genomic sequences contained between two flanking HpaII sites found within 200–2,000 bp from each other. Each fragment on the array is represented by 15 individual probes distributed randomly spatially across the microarray slide. Thus the microarray covers 50,000 CpGs corresponding to 14,000 gene promoters. Signal intensities at each HpaII amplifiable fragment were calculated as a robust (25% trimmed) mean of their component probe-level signal intensities after intensive quality control using analytical pipelines. The $\log_2$(HpaII/MspI) was used as a representative for methylation and analyzed as a continuous variable. For most loci, each fragment was categorized as either methylated, if the centered log HpaII/MspI ratio was less than zero, or hypomethylated if on the other hand the log ratio was greater than zero.

### HELP-tagging for genome-wide methylation analysis of in vitro generated CAFs

The HELP-tagging assay applies massively parallel sequencing to analyze the status of 1.8 million CpGs distributed across the entire genome (*Suzuki et al., 2010*; *Bhattacharyya et al., 2013*). To perform HELP-tagging assays, DNA samples were digested with *Hpa*II and ligated to customized Illumina adapters with a complementary cohesive end. These adapters also contain an EcoP15I site that cuts into the adjacent sequence 27 bp away, allowing us to polish that end and ligate the other Illumina adapter for library generation by PCR. The presence of the CCGG and EcoP15I sequences at the ends of the reads allowed us to remove spurious sequences. Prior to sequencing, we performed qRT-PCR with primers that measure the proportion of adapter dimer complexes in the library, usually a very small proportion (<5%) of the total library. Following sequencing, we removed low quality or unmapped reads, piled up reads on each locus and created an output for each locus in terms of read frequency. We normalized the *Hpa*II signal with that of the deeply-sequenced *Msp*I profiles, as performed previously (*Suzuki et al., 2010*; *Wu et al., 2013*). Results were generated using the WASP system and linked to a local mirror of the UCSC Genome Browser for visualization.

HELP-tagging data were analyzed using an automated pipeline, as described previously (*Suzuki et al., 2010*; *Wu et al., 2013*). Loci were defined in a continuous variable model, given the quantitative nature of this and comparable published assays (*Ball et al., 2009*). Methylation values were depicted from a range of 0 to 100, with 0 representing fully methylated to 100 representing fully hypomethylated loci.

## Whole-genome hydroxymethylation analysis by oxidative bisulfite sequencing

One microgram genomic DNA from MSC and CAF cells were sonicated to 100–400 bp by Bioruptor, and 0.5% (w/w) sequencing spike-in control DNA was added thereafter and purified by Ampure XP beads. Spike-in controls were added to the adapted library. Half of the library was subjected to oxidation reaction following the manufacturer's protocol (Cambridge Epigenetix). Both oxidized and nonoxidized samples were then treated with bisulphite conversion reagent. The final PCR was performed according to the manufacturer's guide using 10 cycles of amplification, purified, and sequenced at the Einstein Epigenomics Facility. Bismark was used to map the sample reads and make methylation calls. At every base location, the 5-mC percentage was estimated by the ratio of nonconverted CpG bases to the total number of bases using the pileup of reads from the OxBS sample. For the estimation of 5-hmC, both BS and OxBS samples were analyzed, and an estimate for the percentage of 5-hmC methylation was calculated by the difference between BS and OXBS conversions.

Since 5-hmC is a less frequent modification, for further stringency in measuring the difference in ratios, we used Fisher's exact test of proportions, using the number of converted and nonconverted reads in the BS and OxBS samples, and selected sites that have a p-value<0.05. 5-mC sites were calculated by the ratio of nonconverted bases to total bases in the OxBS sample with a biologically influenced threshold of 50%.

x.hmc.caf=counts.msccaf[caf.hmc>msc.hmc and caf.hmc >= 0.75 and caf.mc <msc.mc,]

x.hmc.msc=counts.msccaf[caf.hmc<msc.hmc and msc.hmc >= 0.75 and caf.mc >msc.mc,]

CAF hits are the loci where CAF has a higher HMC score than MSC, has an HMC score of at least. 75, and has a lower MC score than MSC.

MSC hits are the loci that have a higher hmc score of at least. 75, and a lower MC score.

To compare hydroxymethylation between cancer and control samples, we used Fisher's test and adjusted for multiple comparisons through the Benjamini-Hochberg procedure.

## Quantitative DNA methylation analysis by MassArray epityping

Validation of HELP microarray findings was carried out by MALDI-TOF mass spectrometry using EpiTyper by MassArray (Sequenom, CA) on bisulfite-converted DNA, as previously described (*Alvarez et al., 2011*; *Figueroa et al., 2008*; *Figueroa et al., 2009*). MassArray primers were designed to cover the flanking *Hpa*II sites for a given locus, as well as any other *Hpa*II sites found up to 2,000 bp upstream of the downstream site and up to 2,000 bp downstream of the upstream site, in order to cover all possible alternative sites of digestion.

### Gene expression microarrays

Gene expression data were obtained using Affymetrix Human Genome U133A 2.0 or Plus2 GeneChips; mRNA isolation, labeling, hybridization, and quality control were carried out as described before (*Alvarez et al., 2011*). Raw data were processed using the Robust Multi-Averaging (RMA) algorithm and Affymetrix Expression Console software. Data are available in the NCBI Gene Expression Omnibus database (GSE101082).

### Microarray data analysis

Unsupervised clustering of HELP and gene expression data by principal component analysis was performed with the use of R 2.8.2 statistical software. Supervised analysis of the methylation data were carried out with a moderated *t*-test with Benjamini-Hochberg correction with a significance level of *P* less than. 05 and an absolute difference in methylation greater than 1.5 between the means of the two populations (eg, MSCs vs. CAFs) to increase the likelihood of detecting biologically significant changes in methylation levels.

## Gene network and gene ontology analysis

Ingenuity Pathway Analysis software and the Database for Annotation, Visualization and Integrated Discovery were used to carry out network composition analyses.

## In vitro generation of CAFs

PANC-1 cells were obtained from ATCC (verified by STR authentication) and were grown in DMEM (Life technologies) + 10% heat-inactivated FBS culture medium and were mycoplasma free. Conditioned medium from pancreatic cancer cell conditioned media (CM) was harvested after 16 hr and centrifuged at 3,000 rpm for 5 min and supernatant was passed through Millipore sterile 50 mL filtration system with 0.45 μm polyvinylidene difluoride membrane. Human mesenchymal stem cells (hMSCs) were exposed to fresh pancreatic cancer cell CM continuously, with the medium changed every third day for the entire 21 day time period.

## Cell lines and reagents

Primary cultures of cancer associated fibroblasts (CAFs) were established from excess tissues of surgically resected pancreatic cancers at the Johns Hopkins Hospital. The excess tissues were obtained delinked from direct patient identifiers, and primary (i.e., non-immortalized) CAFs established by passaging in vitro, as previously described (*Walter et al., 2010*). All CAFs were used at early passage numbers (passages 3–6), and absence of neoplastic epithelium was confirmed by absence of cytokeratin 19 transcripts by qRT-PCR (*data not shown*). For controls, dermal fibroblasts (Hdf), human skin fibroblasts (Hsf) and hepatic stellate (Hst) cells were obtained from ATCC. The human telomerase reverse transcriptase (hTERT) immortalized CAF line, CAF19, was a kind gift from Dr. Michael Goggins at Johns Hopkins University (*Yu et al., 2012*). The human pancreatic ductal adenocarcinoma cell line Pa03C (*Jones et al., 2009*), generated from a liver metastasis, was maintained in DMEM complete media supplemented with 10% FBS and 1% penicillin-streptomycin under mycoplasma free conditions. Mouse pancreatic cancer KPC cells were obtained from Dr Batra (*Torres et al., 2013*). LDH inhibitor FX11 was obtained from Calbiochem and used as previously (*Le et al., 2010*).

## RNA extraction and quantitative real-time reverse transcription PCR

Total RNA was extracted using RNAeasy Mini Kit (Qiagen, Valencia, CA). RNA was reverse transcribed using the TaqMan One-Step RT-PCR Master Mix Reagents Kit (Applied Biosystems, Foster City, CA). Quantitative RT-PCR was carried out using a pre-designed gene expression assay for *CXCR4* (Applied Biosystems) on a StepOnePlus Real-Time PCR System. Relative fold expression was determined and normalized to *GAPDH* (Applied Biosytems) levels using the 2(-ΔΔCT) method (*Schmittgen and Livak, 2008*).

## Transfection of small interfering RNA (siRNA)

De novo (i.e., MSC-derived) CAFs and the immortalized CAF19 cells were seeded and allowed to adhere overnight in 6-well culture plates at a density of $2 \times 10^5$ cells/well. Following overnight incubation, cells were transfected with 50 nmol/L siRNA targeting CXCR4 (Dharmacon Technologies, Thermo Fisher Scientific, Lafayette, CO) or non-targeting control siRNA (Dharmacon Technologies) using DharmaFECT four transfection reagent. At 24 hr post-transfection, cells were plated for co-culture invasion assays.

## Transwell coculture invasion assay

8 μm pore size inserts were coated with 100 μL Matrigel (1:40 Matrigel: PBS solution) (BD Biosciences, San Jose, CA) and allowed to solidify in a notched 24-well culture plate overnight. De novo CAFs or CAF19 cells were then plated in the bottom chamber at a density of $5 \times 10^4$ cells/well and allowed to adhere overnight. The media was then replaced with DMEM containing 1% FBS and Pa03C or PANC-1 cells were suspended in DMEM containing 0.5% FBS and seeded at $5 \times 10^4$/well in the top (notched insert) chamber. Following 48 hr incubation, the assay was terminated and cells migrating to the underside of the insert were fixed in ethanol and stained with 0.25% crystal violet solution. Each condition was performed in triplicate. Invasion assays were also performed in the

presence of the CXCR4 antagonist, AMD3100 (Sigma Aldrich, St. Louis, MO), which was added to the lower chamber.

## Analysis of global DNA methylation and hydroxymethylation by mass spectrophotometer

Genomic DNA was hydrolyzed by DNA Degradase Plus (Zymo Research, CA, USA) according to the manufacturer's instructions. Digested DNA was injected onto a UPLC Zorbax Eclipse Plus C18 RRHD column (Agilent Technologies, CA). The analytes were separated by gradient elution using 5% methanol/0.1% formic acid (mobile phase A) and 100% methanol (mobile phase B) at a flow rate of 0.25 ml/min. Mobile phase B was increased from 0% to 3% in 5 min, to 80% in 0.5 min, kept at 80% for 2 min then switched to initial conditions in 2.5 min. The effluent from the column was directed to the Agilent 6490 Triple Quadrupole mass spectrometer (Agilent Technologies, CA). The following transitions were monitored: m/z 228.1 - > 112.1 (C); m/z 242.1- > 126.1 (5mC) and m/z 258.1- > 142.1 (5hmC).

Calibration solutions with varying amounts of 5hmC (0–3%), 5mC (0–10%) and fixed amount of C, were also analyzed together with the samples. The solutions were prepared from a 200 bp DNA standards containing 57 cytosines which are homogeneous for C, 5hmC or 5mC. Calibration plots of % 5hmC or %5mC vs MRM Response ratio were constructed based on the data obtained. %5hmC is obtained from the ratio of [5hmC]/[5hmC]+[C]. Response ratio is the response peak area for 5hmC or 5mC divided by the combined peak areas of 5hmC, 5mC and C. The % 5hmC or 5mC in the samples were determined from the calibration plots.

## TET enzymatic activity

Tet activity in nuclear cell lysates was assessed by Epigenase 5mC-Hydroxylase TET Activity/Inhibition Assay Kit (Colorimetric, Epigentek). The kit contains cofactors needed for Tet activity in vitro (Ascorbic Acid, aKG and FeNH4SO4) and assesses the amount of active TET enzyme in cell lysates based on efficiency of conversion of 5mC to 5hmC.

## Immunohistochemistry

Tissues/cells were fixed in 10% buffered formalin, embedded in paraffin and sectioned using a microtome and mounted onto glass slides. The slides were incubated at 60 C for an hour to melt the paraffin, followed by dehydrating them through gradients of ethanol (70, 80, 90% and 100%) and 100% xylene. The samples were then treated with antigen unmasking solution (Dako pharma) followed by permeabilization with 0.3% H2O2 and blocked using blocking buffer (5% donkey serum and 2% BSA). Samples were then incubated overnight in the primary antibody prepared in the blocking buffer followed an incubation with appropriate HRP conjugated secondary antibody. Color development was achieved by treating the samples with diaminobenzidine (DAB) and counterstaining performed using harris hematoxylene (Dako pharma). The samples were then passaged through alcohol grades and xylene to dehydrate them, mounted using permount solution (fisher scientific) and allowed to dry overnight before the image analysis.

## In vivo experiments with TET KO mouse stroma coinjection

TET2 Knockout (KO) and wild type (WT) C57B6 mice were euthanized according to protocol and their femur bones were harvested. The bone marrow was flushed and the resulting cells were grown to 30–40% cell density. Once the cells were adhered to the culture flask surface, the WT and KO storm cells were divided into two groups. One group was treated with media conditioned with the KPC cells and the control group was treated with plain culture medium every other day for two weeks. The resulting cells were used for downstream experiments. KPC cells (5 million/mouse) were injected along with TET2 KO stromal or WT stromal cells (1 million/mouse) into NOD-scid IL2R-gamma null (NSG) immuno-deficient mice and then followed for tumor measurements. Mice were sacrificed at end of experiment and tumors used for immunohistochemistry.

## Stable metabolite isotope analysis using GC-MS

*Metabolic extraction* MSCs were cultured with 10 mM $^{13}C_3$ lactate (Cambridge Isotope Labs) for 48 hr. Spent medium was removed, and cells were washed with ice-cold PBS. Cells were then quenched

with 400 µl methanol and 400 µl water containing 1 µg norvaline. Cells were scraped, washed with 800 µl ice-cold choloform, vortexed at 4°C for 30 min and centrifuged at 7,300 rpm for 10 min at 4°C. The aqueous portion was then collected and stored at −80°C until further analysis.

*Derivatization* Samples were first dried and dissolved in 30 µl of 2% methoxyamine hydrochloride in pyridine (Pierce) prior to sonication for 15 min. Samples were then kept at 37°C for 2 hr, and transferred to 55°C for 1 hr following the addition of 45 µl MBTSTFA+1% TBDMCS (Pierce).

*GC-MS measurements* Analysis was done using an Agilent 6890 GC equipped with a 30 m Rtc-5 capillary column connected to an Agilent 5975B MS operating under electron impact ionization at 70 eV. Samples were injected at 1 µl and 270°C in splitless mode, and helium was used as the carrier gas at 1 ml/min. The heating cycle for the GC oven was as follows: 100°C for 3 min, followed by 300°C at 5°C/min temperature increase, for a total run time of 48 min per sample. Data was acquired in scan mode and the integrated signal of all potentially labeled ions was normalized by the norvaline signal and used to calculate the abundance of relative metabolites. The mass isotopomer distribution was obtained by dividing the signal of each isotopomer by the sum of all isotopomer signals and corrected for natural abundance. These stable isotope tracer analysis protocols are established and described in detail in our previous studies (*Achreja et al., 2017*; *Zhao et al., 2018*).

## Nuclear protein extraction and In vitro TET enzymatic activity analysis

Cells were treated with various conditions and nuclear protein was then isolated from cells using the EpiQuik nuclear extraction kit (Epigentek Group Inc), according to the manufacturer's instructions. TET enzymatic activity was measured by using the ELISA-based Epigenase 5mC Hydroxylase TET Activity/Inhibition Assay Kit (Fluorometric) according to manufacturer's instructions. This technique relies on the conversion of methylated products at the bottom of the wells to hydroxymethylated products by the TET enzyme present in the nuclear extract. Thus the amount of hydroxymethylated products formed is a measure of the TET activity of the nuclear extract harvested from the cells being tested.

## Acknowledgements

TB was supported by The Einstein Training Program in Stem Cell Research from the Empire State Stem Cell Fund through New York State Department of Health Contract C34874GG. DVA was supported by a CA200561 T32 training grant. DN is supported by grants from NCI R01CA227622, R01CA222251, and R01CA204969

## Additional information

### Funding

| Funder | Grant reference number | Author |
| --- | --- | --- |
| Albert Einstein College of Medicine | Einstein Training Program in Stem Cell Research C34874GG | Tushar D Bhagat |
| National Cancer Institute | T32 CA200561 | Dagny Von Ahrens |
| National Cancer Institute | R01CA227622 | Deepak Nagrath |
| National Cancer Institute | R01CA222251 | Deepak Nagrath |
| National Cancer Institute | R01CA204969 | Deepak Nagrath |
| National Cancer Institute | R01CA218004 | Anirban Maitra |
| National Cancer Institute | P01CA117969 | Anirban Maitra |
| National Cancer Institute | R01CA220236 | Anirban Maitra |
| National Cancer Institute | U24CA224020 | Anirban Maitra |

The funders had no role in study design, data collection and interpretation, or the decision to submit the work for publication.

## Author contributions
Tushar D Bhagat, Conceptualization, Data curation, Methodology, Project administration; Dagny Von Ahrens, Yiyu Zou, Joelle Baddour, Hongyun Zhao, Lifeng Yang, Gaurav S Choudhary, Beamon Agarwal, Debabrata Banerjee, Data curation, Investigation, Methodology; Meelad Dawlaty, Resources, Data curation; Abhinav Achreja, Data curation, Formal analysis, Investigation, Methodology; Brijesh Patel, Data curation, Formal analysis, Investigation; Changsoo Kwak, Sanchari Bhattacharyya, Srabani Sahu, Matthias Bartenstein, Orsi Giricz, Data curation, Investigation; Shanisha Gordon-Mitchell, Resources, Data curation, Investigation, Methodology; Srinivas Aluri, Data curation, Funding acquisition, Investigation; Prafulla Bhagat, Data curation; Investigation; Methodology; Yiting Yu, Kith Pradhan, Formal analysis, Investigation, Methodology; Masako Suzuki, Resources, Formal analysis, Investigation; Davendra Sohal, Formal analysis, Methodology; Sonal Gupta, Resources, Investigation, Methodology; Paola A Guerrero, Resources, Data curation, Investigation; Surinder Batra, Michael Goggins, Resources, Investigation; Ulrich Steidl, Formal analysis, Investigation, Writing—review and editing; John Greally, Data curation, Formal analysis; Deepak Nagrath, Data curation, Formal analysis, Project administration, Writing—review and editing; Anirban Maitra, Conceptualization, Formal analysis, Project administration, Writing—review and editing; Amit Verma, Conceptualization, Formal analysis, Funding acquisition, Writing—original draft, Project administration, Writing—review and editing

## Author ORCIDs
Tushar D Bhagat (ID) https://orcid.org/0000-0002-4527-5505
Matthias Bartenstein (ID) http://orcid.org/0000-0002-0908-770X
John Greally (ID) http://orcid.org/0000-0001-6069-7960
Deepak Nagrath (ID) https://orcid.org/0000-0002-8999-2282
Amit Verma (ID) https://orcid.org/0000-0001-7592-7693

## Ethics
Animal experimentation: This study was performed in strict accordance with the recommendations in the Guide for the Care and Use of Laboratory Animals of the National Institutes of Health. All of the animals were handled according to approved institutional animal care and use committee (IACUC no. 20181208) protocols of the Albert Einstein College of Medicine.

## Decision letter and Author response
Decision letter https://doi.org/10.7554/eLife.50663.018
Author response https://doi.org/10.7554/eLife.50663.019

# Additional files
## Supplementary files
• Supplementary file 1. Pathways that are overexpressed and demethylated in pancreatic cancer associated fibroblasts.
DOI: https://doi.org/10.7554/eLife.50663.012

• Supplementary file 2. Common sets of genes epigenetically altered in primary and de novo generated CAFs.
DOI: https://doi.org/10.7554/eLife.50663.013

• Supplementary file 3. Differentially expressed transcripts after CXCR knockdown in CAF cells.
DOI: https://doi.org/10.7554/eLife.50663.014

## Data availability
Sequencing data have been deposited in GEO under accession code GSE135218.

The following dataset was generated:

| Author(s) | Year | Dataset title | Dataset URL | Database and Identifier |
|---|---|---|---|---|
| Verma A, Pradhan | 2019 | Lactate-mediated Epigenetic | https://www.ncbi.nlm. | NCBI Gene |

| K | Reprogramming Regulates Formation of Human Pancreatic Cancer-associated Fibroblasts | nih.gov/geo/query/acc. cgi?acc=GSE135218 | Expression Omnibus, GSE135218 |
| --- | --- | --- | --- |

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
