## [Decision Letter]

**Acceptance summary:**

The paper describes widespread epigenetic changes, particularly broadly reduced cytosine methylation, in cancer associated fibroblasts from pancreatic ductal adenocarcinoma (PDAC). These changes are caused by enhanced activity of TET methylcytosine hydroxylases, and result in enhanced expression of CXCR4, which promotes cancer cell invasiveness. Metabolic experiments indicate that cultured PDAC cells secrete lactate, which is taken up by cultured CAFs and used to maintain a large intracellular alpha-ketoglutarate pool, which promotes TET activity. The authors conclude that cancer cell lactate release has a paracrine effect to reprogram gene expression in stromal cells. With increasing appreciation for lactate's involvement in tumor metabolism beyond its presence as a by-product of excessive glycolysis, these new findings are timely and will be interesting to investigators in cancer metabolism and the tumor microenvironment.

**Decision letter after peer review:**

[Editors’ note: a previous version of this study was rejected after peer review, but the authors submitted for reconsideration. The first decision letter after peer review is shown below.]

Thank you for submitting your work entitled "Paracrine Lactate Mediated Epigenetic Reprogramming Regulates Formation of Pancreatic Cancer-associated Fibroblasts" for consideration by *eLife*. Your article has been reviewed by three peer reviewers, one of whom is a member of our Board of Reviewing Editors, and the evaluation has been overseen by a Senior Editor. The reviewers have opted to remain anonymous.

Our decision has been reached after consultation between the reviewers. Based on these discussions and the individual reviews below, we regret to inform you that your work will not be considered further for publication in *eLife*. Although the reviewers found some aspects of the study interesting, there were concerns about conceptual novelty and methodological rigor that will make it difficult to revise the manuscript within the usual 2 month time frame recommended for revision of *eLife* papers. Given that the reviewers found the area of research to be interesting, should you be able to thoroughly address the critiques, the editorial team would consider a revised paper as a new submission.

Reviewer #1:

These authors studied the role of epigenetic reprogramming in cancer-associated fibroblasts (CAFs) from pancreatic ductal adenocarcinoma (PDAC). They find near-global DNA hypomethylation both in primary CAFs and in CAFs generated 'de novo' from mesenchymal stem cells cultivated in medium conditioned by PDAC cells. Hypomethylation was associated with widespread changes in gene expression, including the induction of numerous signaling genes. One such hypomethylated/overexpressed gene in CAFs was CXCR4, and siRNA silencing of this gene in CAFs was sufficient to suppress PDAC cell invasiveness. The authors then use 13C-base isotope tracing to show that lactate, a product secreted by PDAC cells, can be taken up by CAFs and used to supply the TCA cycle. 13C-lactate contributes to part of the CAF aKG pool, and this appeared to be associated with function of TET, an aKG-dependent enzyme, because inhibiting LDH suppressed TET activity. TET inactivation in MSCs also led to reduced production of CAFs and reduced xenograft growth in vivo. The authors conclude that lactate metabolism enables epigenetic reprogramming in CAFs, and that this in turn supports PDAC cell invasion and growth.

This area of work is potentially interesting because relatively little is known about metabolic cross-talk between cell types within the tumor microenvironment. The finding that TET is required for the MSC-to-CAF conversion is also interesting. However, there are several significant conceptual and methodological flaws with the paper in its current form.

1) In Figure 1, epigenetic marks are compared between CAFs and control stromal cells. This is problematic because the CAFs were generated from primary tumor samples and used at low passage, while the controls were purchased from ATCC and no information about passage number is provided. Given the known, potentially widespread epigenetic changes that occur in culture, the large changes reported in the figure could be due to culture-related artifacts. The type of culture medium, which is not described in detail for the cell lines used in the experiment, can also influence these marks. Overall, it is not clear that the authors are studying relevant changes in fibroblast biology.

2) Along these lines, with some 75% of the genome displaying altered cytosine methylation in CAFs, there is remarkably little overlap between promoters affected in primary and de novo-produced CAFs (Figure 2). This raises concern that the de novo CAFs are not particularly good models with which to study CAF epigenetics.

3) In the experiments to manipulate CXCR4, only a single siRNA is used, silencing of mRNA levels does not appear very effective (only about 50% silencing in Figure 2D), no western blots are shown to document reduction in protein levels, and no attempts are made to reverse the effects (e.g. by expressing a non-silenced allele of CXCR4), making it hard to conclude that the siRNA's suppression of PDAC cell invasion are on target.

4) In the metabolism experiments, the authors provide a high dose of 13C-lactate (10 mM) and culture the cells for 48 hours. Yet only 10% of the aKG pool is labeled. This observation provides strong evidence against the main conclusion of the paper: that lactate is an important source of aKG. Citrate is much more extensively labeled by lactate (40%), suggesting that most aKG arises from another carbon source, probably glutamine.

5) The authors should add lactate to the PDAC-conditioned medium in the LDH inhibitor experiment. This would provide evidence that the inhibitor's effect was on target. It would also be useful to perform this experiment genetically by silencing or mutating LDH in the CAFs – this should prevent lactate from being used as a carbon source in these cells.

6) Do the CAFs have any particular predilection to use lactate relative to the PDAC cells? This is an important question, because in the tumor microenvironment, these cells could presumably be competing for lactate.

Reviewer #2:

The present manuscript examines the links between the classical tumor cell metabolite lactate and CAF epigenetics in PDAC. The paper also zooms in on CXCR4 and TET2 as particular genes whose expression in CAFs is important for PDAC growth. The paper also proposes a mechanism by which lactate controls TET2 via KG. Overall, this paper examines a trendy idea (tumor cell secreted products impacting CAF epigenetics) and is concise and an appealing read, but I have serious concerns about the novelty, the quality, and especially the validity of the mechanism:

1) Figure 3 shows that labeled lactate labels TCA intermediates. This is something that was apparent to metabolism experts for ~ 50 years and shown beyond a shadow of a doubt by the Nat Chem paper from Patti lab.

2) Labeling of KG is small (since most KG comes from glutamine in this [not fully physiologic] media). It is most unlikely that 10% of KG coming from lactate can have any affect on epigenetics. There is no evidence presented for altered KG concentration. It is the concentration that controls TET2 activity.

3) The TET2 activity assay is in some version of lysates, which seemingly should not depend on cellular KG as the substrate (methods are not very complete, just referencing a kit). There is no clear logic connecting these biochemical TET2 measurements to KG or lactate. (Obviously lactate can impact cells many ways: pH, energy, pyruvate, NADH, etc.)

4) P-values are reported for inappropriately small N (even N = 2).

5) It is hard to find concentrations of small molecule inhibitors employed, and there is no dose titration for lactate (used at 10 mM-- higher than reported values in PDAC tumors) or for small molecule tool compounds.

6) siRNA KD seem to be reported for single siRNA (evidence is that 5+ are needed for confirming target selectivity given frequent off-target effects) and without rescue (which is of great importance for proving on-target mechanism); there is no use of more advanced and reliable genetics.

7) There are no experiments actually testing paracrine lactate-mediated epigenetic reprogramming in a physiological context.

In summary, the paper is nicely constructed but limited in conceptual novelty and does not come close to proving the proposed mechanism from lactateKGTethypomethylationCXCR4 expression.

Reviewer #3:

This is an excellent study highlighting the role of metabolism reprograming on epigenetics of stromal cells to control tumor growth. I have a few experiments to bolster their conclusions.

Experiments:

1) The authors suggest that lactate is a fuel to support TCA cycle metabolite generation. I assume lactate becomes pyruvate that enters the mitochondria where it is converted into acetyl-CoA via PDH. The authors should prevent pyruvate entry into the mitochondria by using mitochondrial pyruvate carrier inhibitor UK5099 and examine methylation status.

2) Does cell permeable aKG recapitulate the effects of lactate (2 weeks) on methylation in CAFs?

3) It is possible that the presence of lactate diminishes glycolytic flux, which could diminish one-carbon metabolism (SAM levels). They should measure SAM levels as well as provide ratios of succinate/aKG, 2HG/aKG and fumarate/aKG, which have been shown to inhibit TETs, in CAFs +/-lactate.

4) Can they recapitulate the KPC + TET2 KO CAF tumor experiments with KPC + WT CAFs treated with lactate for 2 weeks (Figure 4G)?

[Editors’ note: what now follows is the decision letter after the authors submitted for further consideration.]

Thank you for submitting your article "Lactate-mediated epigenetic reprogramming regulates formation of pancreatic cancer-associated fibroblasts" for consideration by *eLife*. Your article has been reviewed by three peer reviewers, one of whom is a member of our Board of Reviewing Editors, and the evaluation has been overseen by Anna Akhmanova as the Senior Editor. The reviewers have opted to remain anonymous.

The reviewers have discussed the reviews with one another and the Reviewing Editor has drafted this decision to help you prepare a revised submission.

Summary:

The paper describes widespread epigenetic changes, particularly broadly reduced cytosine methylation, in cancer associated fibroblasts from pancreatic ductal adenocarcinoma (PDAC). These changes are caused by enhanced activity of TET methylcytosine hydroxylases, and result in enhanced expression of CXCR4, which promotes cancer cell invasiveness. Metabolic experiments indicate that cultured PDAC cells secrete lactate, which is taken up by cultured CAFs and used to maintain a large intracellular alpha-ketoglutarate pool, which promotes TET activity. The authors conclude that cancer cell lactate release has a paracrine effect to reprogram gene expression in stromal cells.

Essential revisions:

1) There needs to be more clarity about the assays used to measure TET activity in lysates. The authors are proposing that aKG abundance from the lysate influences the activity reported by this commercial assay. But does the assay buffer contain aKG? If so, it seems unlikely that the relatively small amount of aKG in the lysates could substantially affect TET activity. Please clarify: (i) The source and abundance of aKG in the assay, which is not clear from the methods or the vendor website; and (ii) based on the answer, a clear delineation in a few sentences of reasonable possibilities of how components in the lysate (including aKG) might influence the results of the assay.

2) In the original submission, two reviewers questioned the importance of lactate for formation of aKG when isotope tracing with 13C-lactate resulted only in very low labeling of the aKG pool. The explanation for this result seems to be that lactate-derived synthesis of alanine results in production of unlabeled aKG; therefore observation of labeled aKG underestimates lactate's contribution. If this is the argument, it should be spelled out clearly in the manuscript itself, because the impression is still that lactate makes only a small contribution to aKG. The authors also need to explain why the fractional enrichment in alanine is so different between Figure 3A and Figure 3—figure supplement 1A. The latter was used to make the point about alanine labeling in the rebuttal, but the enrichment in both high and low glucose conditions in this experiment is much higher than in the main figure. Finally, it would help the reader if the "accessory" formation of aKG via alanine synthesis (i.e. that pyruvate's conversion to alanine results in glutamate's conversion to aKG) was clearly illustrated in the model in Figure 3F.

3) Evidence for paracrine lactate exchanges in vivo is still circumstantial. Some of the same phenomena are observed in culture and in vivo, and the evidence for TET2's involvement in vivo is strong. But there is still no compelling evidence that lactate released by cancer cells has anything to do with TET2 activity in CAFs in the tumor microenvironment. Adding direct evidence along these lines would strengthen the paper. Failing that, the authors need to be more circumspect about the role of this mechanism in tumors.

4) The authors should address how they separate the effects of lactate carbon from the effects of pH in epigenetic reprogramming. This could be addressed in the Discussion.

---

## [Author Response]

[Editors’ note: the author responses to the first round of peer review follow.]

Reviewer #1:These authors studied the role of epigenetic reprogramming in cancer-associated fibroblasts (CAFs) from pancreatic ductal adenocarcinoma (PDAC). They find near-global DNA hypomethylation both in primary CAFs and in CAFs generated 'de novo' from mesenchymal stem cells cultivated in medium conditioned by PDAC cells. Hypomethylation was associated with widespread changes in gene expression, including the induction of numerous signaling genes. One such hypomethylated/overexpressed gene in CAFs was CXCR4, and siRNA silencing of this gene in CAFs was sufficient to suppress PDAC cell invasiveness. The authors then use 13C-base isotope tracing to show that lactate, a product secreted by PDAC cells, can be taken up by CAFs and used to supply the TCA cycle. 13C-lactate contributes to part of the CAF aKG pool, and this appeared to be associated with function of TET, an aKG-dependent enzyme, because inhibiting LDH suppressed TET activity. TET inactivation in MSCs also led to reduced production of CAFs and reduced xenograft growth in vivo. The authors conclude that lactate metabolism enables epigenetic reprogramming in CAFs, and that this in turn supports PDAC cell invasion and growth.This area of work is potentially interesting because relatively little is known about metabolic cross-talk between cell types within the tumor microenvironment. The finding that TET is required for the MSC-to-CAF conversion is also interesting. However, there are several significant conceptual and methodological flaws with the paper in its current form.1) In Figure 1, epigenetic marks are compared between CAFs and control stromal cells. This is problematic because the CAFs were generated from primary tumor samples and used at low passage, while the controls were purchased from ATCC and no information about passage number is provided. Given the known, potentially widespread epigenetic changes that occur in culture, the large changes reported in the figure could be due to culture-related artifacts. The type of culture medium, which is not described in detail for the cell lines used in the experiment, can also influence these marks. Overall, it is not clear that the authors are studying relevant changes in fibroblast biology.

The reviewer raises a good point. We used 4 distinct control fibroblasts and stellate cells that were low passage and were converted to DNA/RNA without growing them in vitro. We demonstrated loss of methylation in both primary CAFs as well as de novo generated CAFs. Now we have conducted whole genome hydroxymethylome and methylome analysis of CAFs (new Figure 4) and show that many important CAF related genes gain 5hmC marks. This is the first demonstration of genome wide hydoxymethylation in any tumor microenvironmental study. Additionally, we have also shown increased 5hmC by Mass Spec also. Taken together, all of these data point to reduced methylation and increased 5hmC in cancer associated fibroblasts.

2) Along these lines, with some 75% of the genome displaying altered cytosine methylation in CAFs, there is remarkably little overlap between promoters affected in primary and de novo-produced CAFs (Figure 2). This raises concern that the de novo CAFs are not particularly good models with which to study CAF epigenetics.

The analysis for the primary CAFs and de novo CAFs was done by two different platforms. There were different genomic loci that were represented on the HELP assay arrays that were used accounting for some of the discordance. Based on the reviewer’s question, we now conducted whole genome methylome and 5hmC analysis with oxidative bisulfite sequencing on MSCs and CAFs and demonstrate that virtually every CAF associated gene examined exhibits strong gains in 5hmC marks with concomitant loss of mC marks. These new data demonstrate that the loss of methylation in CAFs is now validated by many different assays.

3) In the experiments to manipulate CXCR4, only a single siRNA is used, silencing of mRNA levels does not appear very effective (only about 50% silencing in Figure 2D), no western blots are shown to document reduction in protein levels, and no attempts are made to reverse the effects (e.g. by expressing a non-silenced allele of CXCR4), making it hard to conclude that the siRNA's suppression of PDAC cell invasion are on target.

We have now used a second set of siRNAs with consistent results. We also show reduction at the protein level with immunoblotting (Figure 2—figure supplement 1). Our new 5hmC analysis also demonstrates acquisition of 5hmC not only at CXCR4 promoter, but also at the CXCR4 enhancer, strengthening the proposed hypothesis. Finally, we have now shown increased CXCR4 expression in the stromal cells in a set of 261 primary pancreatic cancer tissues (TMA) as shown in the new Figure 5.

4) In the metabolism experiments, the authors provide a high dose of 13C-lactate (10 mM) and culture the cells for 48 hours. Yet only 10% of the aKG pool is labeled. This observation provides strong evidence against the main conclusion of the paper: that lactate is an important source of aKG. Citrate is much more extensively labeled by lactate (40%), suggesting that most aKG arises from another carbon source, probably glutamine.

We thank the reviewers for making this observation, and we agree with them to a certain extent. However, the labeling of alpha-ketoglutarate (aKG) is difficult to take at face value as a representation of carbon contribution from any substrate. This is because aKG is a branch metabolite that connects the TCA cycle, anaplerotic glutamine flux and transamination reactions. Further, the conversion of Citrate to aKG, as well as Glutamate to aKG are reversible reactions. In the presence of glutamine and a source for pyruvate, equilibrium favors conversion of both Citrate and Glutamate to aKG. In order to further support our hypothesis, we repeated the 13-carbon labeled lactate experiment, with an additional condition of low glucose and low glutamine to highlight the contribution of lactate as a carbon substrate.

In these experiments as well as the experiments in our manuscript, the 13-carbon contribution from citrate is diluted by the 12-carbon from glutamate. This is also corroborated from the high 13-carbon labeling of alanine shown in Author Response Image 1, where the net production of 13-carbon labeled Alanine from pyruvate (derived from 13-carbon lactate) is accompanied by a net production of aKG from glutamate.

**Author response image 1. respfig1:** Mass Isotopologue Distribution of Alanine from U-13C-Lactate tracing experiments indicate that lactate-derived carbon contributes significantly to de novo alanine synthesis.

The closest representation of carbon contribution from citrate to aKG, therefore, is represented by the total 13-carbon enrichment (mole percent enrichment, MPE) in citrate and aKG. The ration of aKG MPE to citrate MPE will indicate the net carbon contribution from 13-carbon lactate to citrate to aKG (Author response image 2).

**Author response image 2. respfig2:** 

This data strongly supports our hypothesis that lactate is a significant carbon source for aKG, and this is especially evident under low glucose and low glutamine condition where lactate-derived citrate contributes to over 50% of the aKG carbon.

5) The authors should add lactate to the PDAC-conditioned medium in the LDH inhibitor experiment. This would provide evidence that the inhibitor's effect was on target. It would also be useful to perform this experiment genetically by silencing or mutating LDH in the CAFs – this should prevent lactate from being used as a carbon source in these cells.

We have now conducted LDH knockdown on MSC/CAFss and shown that this results in reduced expression of CAF conversion as evident from significantly reduced expression of FSP1, aSMA and VIM (Figure 3—figure supplement 2). Further experiments with exogenous lactate have been included in Figure 3—figure supplement 2.

6) Do the CAFs have any particular predilection to use lactate relative to the PDAC cells? This is an important question, because in the tumor microenvironment, these cells could presumably be competing for lactate.

This is an important point raised by the reviewer. In fact, we have recently shown that lactate secreted from cancer cells is preferably uptaken by CAFs (Nagrath et al., Cell Metab. 2016 Nov 8;24(5):685-700; Targeting Stromal Glutamine Synthetase in Tumors Disrupts Tumor Microenvironment-Regulated Cancer Cell Growth). In this work, we showed that CAFs used more lactate than glucose for acetyl-CoA synthesis.

Reviewer #2:The present manuscript examines the links between the classical tumor cell metabolite lactate and CAF epigenetics in PDAC. The paper also zooms in on CXCR4 and TET2 as particular genes whose expression in CAFs is important for PDAC growth. The paper also proposes a mechanism by which lactate controls TET2 via KG. Overall, this paper examines a trendy idea (tumor cell secreted products impacting CAF epigenetics) and is concise and an appealing read, but I have serious concerns about the novelty, the quality, and especially the validity of the mechanism:1) Figure 3 shows that labeled lactate labels TCA intermediates. This is something that was apparent to metabolism experts for ~ 50 years and shown beyond a shadow of a doubt by the Nat Chem paper from Patti lab.

We agree that it has been shown that lactate can lead to generation of TCA intermediates, but that biochemical observation was not the main message and novelty of our paper. We wanted to explore the molecular mechanisms of how CAFs are generated. Our paper is the first report to show epigenomic alterations during the process of MSCs to CAFs. Our paper is also the first report to point to the contribution of lactate in CAF generation. There have been recent reports on the contribution of lactate and other secreted agents on the “malignant cells”; but there have not been any reports demonstrating that lactate or any other secreted agent can lead reprogramming of MSCs to CAFs within the tumor micro-environment.

2) Labeling of KG is small (since most KG comes from glutamine in this [not fully physiologic] media). It is most unlikely that 10% of KG coming from lactate can have any affect on epigenetics. There is no evidence presented for altered KG concentration. It is the concentration that controls TET2 activity.

We have now shown that exogenous lactate leads to increased intracellular concentrations of aKG in CAFs (Figure 3—figure supplement 2) and that increased amounts of cell permeable aKG leads to increased TET activity (Figure 3—figure supplement 2). Furthermore, as discussed in Reviewer 1, point 4, our new metabolic experiments demonstrate the importance of lactate as a source of aKG (Figure 3—figure supplement 1).

3) The TET2 activity assay is in some version of lysates, which seemingly should not depend on cellular KG as the substrate (methods are not very complete, just referencing a kit). There is no clear logic connecting these biochemical TET2 measurements to KG or lactate. (Obviously lactate can impact cells many ways: pH, energy, pyruvate, NADH, etc.)

The TET activity assay was reliant on nuclear lysates generated from CAFs and MSCs by us. The nuclear lysate included the TET protein as well as other proteins, incl aKG that are present in the lysate. The activity is assessed by an ELISA that relies on ability of TET in converting mC to hmC. We have now added details on this assay to the methods section. We have used this assay in multiple recent papers that show TET activation by ascorbic acid (Shenoy, Verma et al., JCI 2019) and by phosphorylation (Jeong, Verma et al., Cancer Discovery, 2019, In Press).

4) P-values are reported for inappropriately small N (even N = 2).

The two assays that the reviewer is referencing to were de novo generation of CAFs from primary human MSC cultures. These were two independent biological experiments that rely on 21 day culture process in the presence of conditioned media. Due to the number of cells required (many millions) and the use of primary cells, we felt two independent biological replicates will be sufficient for the mass spec analysis of hmC. We have now conducted another 2 MSC to CAF conversion assays with biologically similar results and have shown them in supplementary figures.

5) It is hard to find concentrations of small molecule inhibitors employed, and there is no dose titration for lactate (used at 10 mM-- higher than reported values in PDAC tumors) or for small molecule tool compounds.

We used 10mM dose of Lactate based on published lactate concentrations in numerous studies in the stromal field and also from data generated from patient serum levels (Nature Cell Biology, (20), 597–609 (2018) Cancer-cell-secreted exosomal miR-105 promotes tumor growth through the MYC-dependent metabolic reprogramming of stromal cells; Walenta et al., Cancer Res. 2000 Feb 15;60(4):916-21; Semin Radiat Oncol. 2004 Jul; 14(3):267-74). Furthermore, a new study in e*Life* confirms high lactate levels in vivo in the same range (5mM to 10mM range) in pancreatic cancer and lung cancer models (Sullivan and Muir, e*Life*; https://elifesciences.org/articles/44235). The dose of LDH inhibitor, FX11 (10uM) was identical to the dose used in the original paper describing this specific inhibitor Chi Dang et al., Inhibition of lactate dehydrogenase A induces oxidative stress and inhibits tumor progression PNAS, 107(5), 2037–2042). We have now added these references in the text.

6) siRNA KD seem to be reported for single siRNA (evidence is that 5+ are needed for confirming target selectivity given frequent off-target effects) and without rescue (which is of great importance for proving on-target mechanism); there is no use of more advanced and reliable genetics.

We now show data with second set of siRNAs and also show reduction at the level of protein expression (Figure 2—figure supplement 1). Also, we have now shown 5hmC acquisition and CXCR4 overexpression in a cohort of 250+ primary PDAC sample, in single cell RNA-seq studies as well as in the KPC mouse model (new Figure 5).

7) There are no experiments actually testing paracrine lactate-mediated epigenetic reprogramming in a physiological context.In summary, the paper is nicely constructed but limited in conceptual novelty and does not come close to proving the proposed mechanism from lactateKGTethypomethylationCXCR4 expression.

As mentioned in response to point no 5, we based the lactate concentrations on multiple published studies. In fact, a new study in *eLife* confirms high lactate levels in vivo in the same range (5mM to 10mM range) in pancreatic cancer and lung cancer models (Sullivan and Muir, *eLife*; https://elifesciences.org/articles/44235). We have now shown that TET activity increases in a dose dependent fashion when MSCs are exposed to exogenous lactate (Figure 3—figure supplement 2). We also show the role of TET enzymes in CAF formation in a mouse model in vivo (Figure 5).

Reviewer #3:This is an excellent study highlighting the role of metabolism reprograming on epigenetics of stromal cells to control tumor growth. I have a few experiments to bolster their conclusions.Experiments:1) The authors suggest that lactate is a fuel to support TCA cycle metabolite generation. I assume lactate becomes pyruvate that enters the mitochondria where it is converted into acetyl-CoA via PDH. The authors should prevent pyruvate entry into the mitochondria by using mitochondrial pyruvate carrier inhibitor UK5099 and examine methylation status.

Based on the reviewers suggestion we have performed experiments using pyruvate inhibitor UK5099 and show that this inhibition reduces the TET2 activation as well as 5hmC increase that is seen by exogenous lactate in CAF cells (Figure 3—figure supplement 2).

2) Does cell permeable aKG recapitulate the effects of lactate (2 weeks) on methylation in CAFs?

It has also been shown previously that cell permeable aKG does lead to TET activation and increased 5hmC (Carey et al., 2015). Based on the reviewer’s suggestion, we used a cell permeable aKG and show that TET2 activity does increase with this agent in CAF cells (Figure 3—figure supplement 2).

3) It is possible that the presence of lactate diminishes glycolytic flux, which could diminish one-carbon metabolism (SAM levels). They should measure SAM levels as well as provide ratios of succinate/aKG, 2HG/aKG and fumarate/aKG, which have been shown to inhibit TETs, in CAFs +/-lactate.

The reviewer brings out an excellent observation and we thank them for bringing this to our attention. With regards to lactate reducing glycolytic flux and subsequently affecting SAM cycle, unfortunately, direct measurement of S-adenosyl methionine is difficult with GC-MS techniques used for measuring polar metabolites. However, we looked at serine and glycine levels in the presence of lactate, since glycolytic carbon to SAM cycle is routed through serine and glycine. As the reviewer anticipated, we do find an indication that glycolytic flux to SAM cycle might be decreasing by observing the Serine:Pyruvate abundance ratio (Figure 3—figure supplement 1).

As for the metabolite ratio being indicative of TET activity, we calculated 2-HG:aKG and Fumarate:aKG ratio in media with and without Lactate. The 2-HG signals were low across all the samples, which was expected as 2-HG accumulation is prevalent in the presence of mutated IDH and there is no evidence IDH mutations in these cells. The small amounts of 2-HG detected are presumably from the promiscuous IDH activity. Despite this, the 2-HG:aKG ratio in our observations tend to decrease when lactate is present in the media, indicating higher TET activity.

More interestingly, Fumarate:aKG ratio drops markedly in the presence of lactate, indicating decrease in TET inhibition as observed in the studies by Xiao et al. and Laukka et al. Unfortunately, total succinate levels could not be estimated correctly due to succinate and proline co-eluting in our GC-MS analysis.

Xiao M, Yang H, Xu W, et al. Inhibition of α-KG-dependent histone and DNA demethylases by fumarate and succinate that are accumulated in mutations of FH and SDH tumor suppressors. Genes Dev. 2012;26(12):1326-38.

Laukka, T., Mariani, C.J., Ihantola, T., Cao, J.Z., Hokkanen, J., Kaelin, W.G., Godley, L.A., and Koivunen, P. (2016). Fumarate and Succinate Regulate Expression of Hypoxia-inducible Genes via TET Enzymes. J. Biol. Chem. 291, 4256–4265.

All of these results are now shown in Figure 3—figure supplement 1 of the revised manuscript.

4) Can they recapitulate the KPC + TET2 KO CAF tumor experiments with KPC + WT CAFs treated with lactate for 2 weeks (Figure 4G)?

We appreciate the reviewer’s point. Our aim was to show that MSCs from TET2 KO mice will not be able to form functional CAFs and will not be able to support tumor growth as well as the WT CAFs. Based on the reviewer’s advice we did treat WT CAFs with exogenous lactate. Unfortunately, these primary mouse derived cells did not grow very well in vitro and we did not achieve sufficient numbers to perform the in vivo xenograft studies.

[Editors' note: the author responses to the re-review follow.]

Essential revisions:1) There needs to be more clarity about the assays used to measure TET activity in lysates. The authors are proposing that aKG abundance from the lysate influences the activity reported by this commercial assay. But does the assay buffer contain aKG? If so, it seems unlikely that the relatively small amount of aKG in the lysates could substantially affect TET activity. Please clarify: (i) The source and abundance of aKG in the assay, which is not clear from the methods or the vendor website; and (ii) based on the answer, a clear delineation in a few sentences of reasonable possibilities of how components in the lysate (including aKG) might influence the results of the assay.

We were able to obtain information from the vendor (Epigentek) that the TET activity assay buffer does contain cofactors required for TET enzymatic activity: Ascorbic Acid (1mM), aKG (1mM) and FeNH4SO4 (0.1mM). The assay does not contain TET though and thus relies on TET obtained from the nuclear cell lysates. The assay is also developed very quickly (5 minutes). The assay relies on the conversion of 5mC to 5hmC by TET enzymatic activity. Notwithstanding that the assay contains aKG; we were still able to obtain a significant difference in TET activity between the CAFs and MSCs, suggesting that CAFs potentially contained activated TET that was able to convert mC to hmC in a short period of time. We have now added text about the constituents of the assay and the inferred conclusions in the revised text.

2) In the original submission, two reviewers questioned the importance of lactate for formation of aKG when isotope tracing with 13C-lactate resulted only in very low labeling of the aKG pool. The explanation for this result seems to be that lactate-derived synthesis of alanine results in production of unlabeled aKG; therefore observation of labeled aKG underestimates lactate's contribution. If this is the argument, it should be spelled out clearly in the manuscript itself, because the impression is still that lactate makes only a small contribution to aKG. The authors also need to explain why the fractional enrichment in alanine is so different between Figure 3A and Figure 3—figure supplement 1A. The latter was used to make the point about alanine labeling in the rebuttal, but the enrichment in both high and low glucose conditions in this experiment is much higher than in the main figure. Finally, it would help the reader if the "accessory" formation of aKG via alanine synthesis (i.e. that pyruvate's conversion to alanine results in glutamate's conversion to aKG) was clearly illustrated in the model in Figure 3F.

The reviewer brings up an important point. The fractional enrichment of alanine in the main figure and supplementary figures is disparate due to the different media formulations used. The data in the main figure corresponds to MSCs cultured in aplhaMEM with 13C-lactate, whereas the supplementary figure shows results for MSCs cultured in RPMI with 13C-lactate. Since α-MEM contains unlabeled alanine, the 13C-enrichment of intracellular alanine is diluted and appears less than the corresponding enrichment of alanine in RPMI media, which does not contain alanine. The MSCs were cultured in both α-MEM and RPMI to highlight the lactate-derived synthesis of alanine via GPT that was being masked in α-MEM cultures. In addition, we have amended the schematic to show the accessory formation of aKG via GPT in revised Figure 3A.

3) Evidence for paracrine lactate exchanges in vivo is still circumstantial. Some of the same phenomena are observed in culture and in vivo, and the evidence for TET2's involvement in vivo is strong. But there is still no compelling evidence that lactate released by cancer cells has anything to do with TET2 activity in CAFs in the tumor microenvironment. Adding direct evidence along these lines would strengthen the paper. Failing that, the authors need to be more circumspect about the role of this mechanism in tumors.

Based on the reviewers’ suggestion, we have changed the language in Abstract, Introduction and Discussion in the paper as follows, to alleviate the reviewer’s concerns:

Abstract: We have added “associated” instead of “results” and “seen in” instead of “regulates”

Previous line: Thus, in PDAC, a tumor-mediated lactate flux results in widespread epigenomic reprogramming that regulates CAF formation

Revised line: Thus, in PDAC, a tumor-mediated lactate flux is associated with widespread epigenomic reprogramming that is seen during CAF formation.

In the Introduction: “Importantly, from a mechanistic standpoint, we determine that paracrine lactate secreted by PDAC cells can be incorporated in stromal cells and lead to increased alpha-keto glutarate (aKG). This is associated with activation of the TET demethylase, thus potentially leading to epigenetic reprogramming seen during CAF formation.

We have made similar changes in the Discussion also.

4) The authors should address how they separate the effects of lactate carbon from the effects of pH in epigenetic reprogramming. This could be addressed in the Discussion.

This is an excellent point raised by the reviewer. In fact, we did find one manuscript that showed that low acidic ph leads to increases in 2HG more than aKG (Nadtochiy et al., 2016). 2HG increase generally leads to decreased Tet activity. Thus in our model, a low ph (due to increase lactic acid) should not account for the high Tet activity and consequent decreased 5mC that we see. We have added this in the Discussion also.